# Iron photocatalysis via Brønsted acid-unlocked ligand-to-metal charge transfer

Xiaoyu Jiang[1], Yu Lan [1,2,3] ✉, Yudong Hao[1], Kui Jiang[1], Jing He[1], Jiali Zhu[1], Shiqi Jia[1], Jinshuai Song [1], Shi-Jun Li [1,2] ✉ & Linbin Niu [1,2] ✉

Reforming sustainable 3d-metal-based visible light catalytic platforms for inert bulk chemical activation is highly desirable. Herein, we demonstrate the use of a Brønsted acid to unlock robust and practical iron ligand-to-metal charge transfer (LMCT) photocatalysis for the activation of multifarious inert haloalkylcarboxylates ($C_nX_mCOO^-$, X = F or Cl) to produce $C_nX_m$ radicals. This process enables the fluoro-polyhaloalkylation of non-activated alkenes by combining easily available Selectfluor as a fluorine source. Valuable alkyl fluorides including potential drug molecules can be easily obtained through this protocol. Mechanistic studies indicate that the real light-harvesting species may derive from the in situ-assembly of $Fe^{3+}$, $C_nX_mCOO^-$, $H^+$, and acetonitrile solvent, in which the Brønsted acid indeed increases the efficiency of LMCT between the iron center and $C_nX_mCOO^-$ via hydrogen-bond interactions. We anticipate that this Brønsted acid-unlocked iron LMCT platform would be an intriguing sustainable option to execute the activation of inert compounds.

Developing sustainable catalytic strategies for radical generation and transformation from cheap and particularly inert bulk chemicals is what radical chemists are continually pursuing[1,2]. With the renaissance of photochemistry[3–12], photo-induced ligand-to-metal charge transfer (LMCT) of photochemically active metal complex has been served as a robust tool for the generation of open-shell radical species. Up to now, exploration of the ligands for photochemically active metal complexes have mainly focused on $Cl^-$, $N_3^-$, alcohols, and easily-activated aliphatic carboxylic acids[13–21]. Expanding the ligand scope of LMCT to more inert compounds is one of the most important tasks for the development of photo-induced LMCT chemistry. Recently, UV or purple light-induced copper LMCT strategies for the decarboxylative functionalization of stable benzoic acid (PhCOO^-, $E_{1/2}^{ox}$ = 1.4 V versus saturated calomel electrode (SCE)) were reported by Ritter and MacMillan[22–26]. Notably, there are still numerous cheap bulk chemicals with extremely high oxidative potentials such as trifluoroacetate ($E_{1/2}^{ox}$ > 2.4 V versus SCE)[27,28], making the activation and transformation via visible light-induced

LMCT quite challenging[29,30]. Therefore, enhancing the LMCT reactivity of extremely inert bulk chemicals as ligands for radical production is highly desirable (Fig. 1a).

Iron as one of the cheapest and most abundant metals[31,32], has displayed impressive photochemical activities in photochemistry[33–47], whose LMCT catalytic ability for radical species production is the ideal alternative to the single electron transfer (SET) capacity of the excited noble-metal ruthenium and iridium polypyridyl complexes. With the urgent demand for sustainable chemistry, we anticipate the development of a robust iron LMCT catalytic platform that can be applied for inert bulk chemical activation and value-added molecular skeleton construction. Considering the privileged status of fluorine and/or fluorine-containing groups in new pharmaceuticals, agrochemicals, functional materials, and organic synthesis[48–58], we hope to utilize the ubiquitous $C_nX_mCOO^-$ (X = F or Cl) as a $C_nX_m$ radical source[59–66] and alkene as a synthetic linker to perform visible light-induced iron-catalyzed fluoro-polyhaloalkylation with an easily available and commercial fluorination reagent. This would provide a low-cost and

[1]College of Chemistry, and Pingyuan Laboratory, Zhengzhou University, Zhengzhou, Henan, PR China. [2]State Key Laboratory of Antiviral Drugs, Pingyuan Laboratory, Henan Normal University, Xinxiang, Henan, PR China. [3]School of Chemistry and Chemical Engineering, Chongqing Key Laboratory of Chemical Theory and Mechanism, Chongqing University, Chongqing, PR China. ✉e-mail: lanyu@cqu.edu.cn; lishijunzong@zzu.edu.cn; nlb@zzu.edu.cn

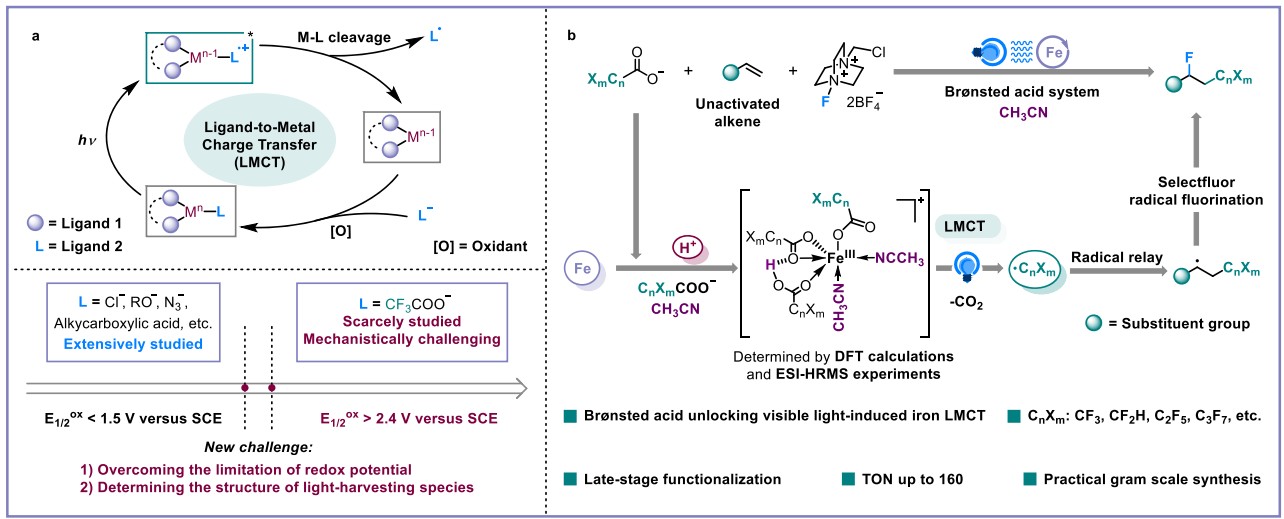

**Fig. 1 | Photo-induced metal-based LMCT for radical production and transformation. a** The state-of-the-art of photo-induced LMCT. **b** This work: Brønsted acid unlocking iron LMCT for fluoro-polyhaloalkylation of alkenes. DFT density functional theory, ESI-HRMS electrospray ionization high-resolution mass spectrometry.

universal method for the synthesis of valuable haloalkanes[67,68]. However, the negative inductive effects of halogens seriously weaken the coordination ability and electron density of carboxyl groups, making it very difficult to effectively assemble iron and $C_nX_mCOO^-$-based light-harvesting species via LMCT to generate $C_nX_m$ radicals under visible light irradiation.

To address this intricate issue, through detailed mechanistic studies, we herein disclose the use of a Brønsted acid to unlock a practical iron LMCT catalytic platform for fluoro-polyhaloalkylation of non-activated alkenes with abundant $C_nX_mCOO^-$ as the $C_nX_m$ source and electrophilic Selectfluor as the fluorination reagent (Fig. 1b). A variety of non-activated alkenes can be successfully converted into alkyl fluorides with excellent regioselectivity and moderate to good yields. Notably, the features of late-stage fluoro-polyhaloalkylation of complex drug-like molecules, gram-scale synthesis, and low loading amount of iron catalyst (Turnover Number, TON: 160), show potential application prospects in pharmaceutical discovery and synthetic chemistry.

## Results

### Reaction development

To develop a general Brønsted acid-unlocked iron LMCT protocol for the activation of $C_nX_mCOO^-$ to form $C_nX_m$ radicals, we proposed that balancing the concentration of Brønsted acid and ligand $C_nX_mCOO^-$ is vital. Therefore, we expect to adopt the in situ generation strategy for the release of the Brønsted acid and $C_nX_mCOO^-$ (Fig. 2a, Supplementary Figs. 13 and 14, 18–20). Through $^{19}F$ NMR identification, it was found that the combination of trifluoroacetic anhydride (TFAA) and the OH nucleophile-containing isopropanol could afford the strong Brønsted acid trifluoroacetic acid (TFA). When isopropanol was replaced by oxydibenzene, the possible trifluoroacetates (**3** and **4**) and by-product **5** were detected by electrospray ionization-high resolution mass spectros (ESI-HRMS). We thought that **4** was generated from the nucleophilic substitution between **3** and $CH_3CN$. This process would accomplish the regeneration of oxydibenzene, and **4** might eventually deliver **5** with the release of TFA. Then, we decided to choose cheap TFAA, isopropanol, oxydibenzene, and $CH_3CN$ to construct a suitable Brønsted acid and trifluoroacetate reaction system. Following our continuous effort, we identified suitable and manageable conditions that directly employed commercial $Fe(acac)_3$ as the catalyst and Selectfluor as the radical fluorination reagent to successfully achieve the fluorotrifluoromethylation of non-activated alkenes using a Brønsted acid and trifluoroacetate reaction system and blue LED

irradiation (Fig. 2b–i, Supplementary Tables 4 and 5). When the oxydibenzene was removed from the standard conditions, only 15% yield of desired fluorotrifluoromethylation product was detected, accompanied by the complete consumption of alkene **6**. After ESI-HRMS detection, the possible aminotrifluoromethylation and fluoroamination of alkenes (**8** and **9**) were considered as the major transformations (Fig. 2b–ii, Supplementary Figs. 16, 21, and 22). Similarly, in the absence of oxydibenzene, utilizing N-Fluorobenzenesulfonimide as a weaker electrophilic fluorination reagent than Selectfluor also prefers the C–N bond formation owing to the polarity matching effect between the *N* radical intermediate and non-activated alkenes (Supplementary Figs. 17 and 23). Based on previous reports[69,70], the generation of **9** involves the radical propagation of *N* radical cation **10** with an alkene, and thus the second role of oxydibenzene might serve as the redox buffer to timely quench the electrophilic *N* radical cation from the Selectfluor (Fig. 2b–iii). Significantly, compared with the direct TFA system and TFAA-oxydibenzene conditions (without isopropanol), the faster initial reaction rate and shorter induction period of our standard conditions evidence the necessity and advantages of our balanced Brønsted acid and ligand $C_nX_mCOO^-$ system (Fig. 2c, Supplementary Fig. 15).

### Mechanistic studies

Detailed mechanistic studies were performed to identify the real light-harvesting species. During the investigation of the iron salt catalyst, we found that $FeCl_3$ and $Fe(CF_3COO)_3$ behaved satisfying catalytic ability for alkene fluorotrifluoromethylation (Supplementary Table 2, Entries 1 and 2). Therefore, we proposed that these iron salts could be transformed into iron and $CF_3COO^-$-based light-harvesting species in the constructed Brønsted acid ($H^+$) and trifluoroacetate ($CF_3COO^-$) system. This proposal was evidenced by detailed UV–Vis studies (Supplementary Fig. 26). Further stoichiometric experiments revealed that under the blue LED irradiation, $Fe(CF_3COO)_3$ can be unlocked and release $CF_3$ radicals only in the presence of a strong Brønsted acid (Fig. 2d, Entries 1–4). Weak Brønsted acids or exogenous $CF_3COONa$ did not promote the activation of $Fe(CF_3COO)_3$ for the fluorotrifluoromethylation of alkene (Entries 5–7). The addition of a Brønsted acid successfully avoided the need for UV light for the transformation of $Fe(CF_3COO)_3$ (Entries 8 and 9). The results of UV–Vis studies also indicated that the photodecomposition of $Fe(CF_3COO)_3$ would occur only when $Fe(CF_3COO)_3$ was subjected to acidic conditions. In the absence of a Brønsted acid, extra $CF_3COONa$ could still not assist with the visible light-induced LMCT of $Fe(CF_3COO)_3$ (Fig. 2e). Besides, using $Fe(acac)_3$

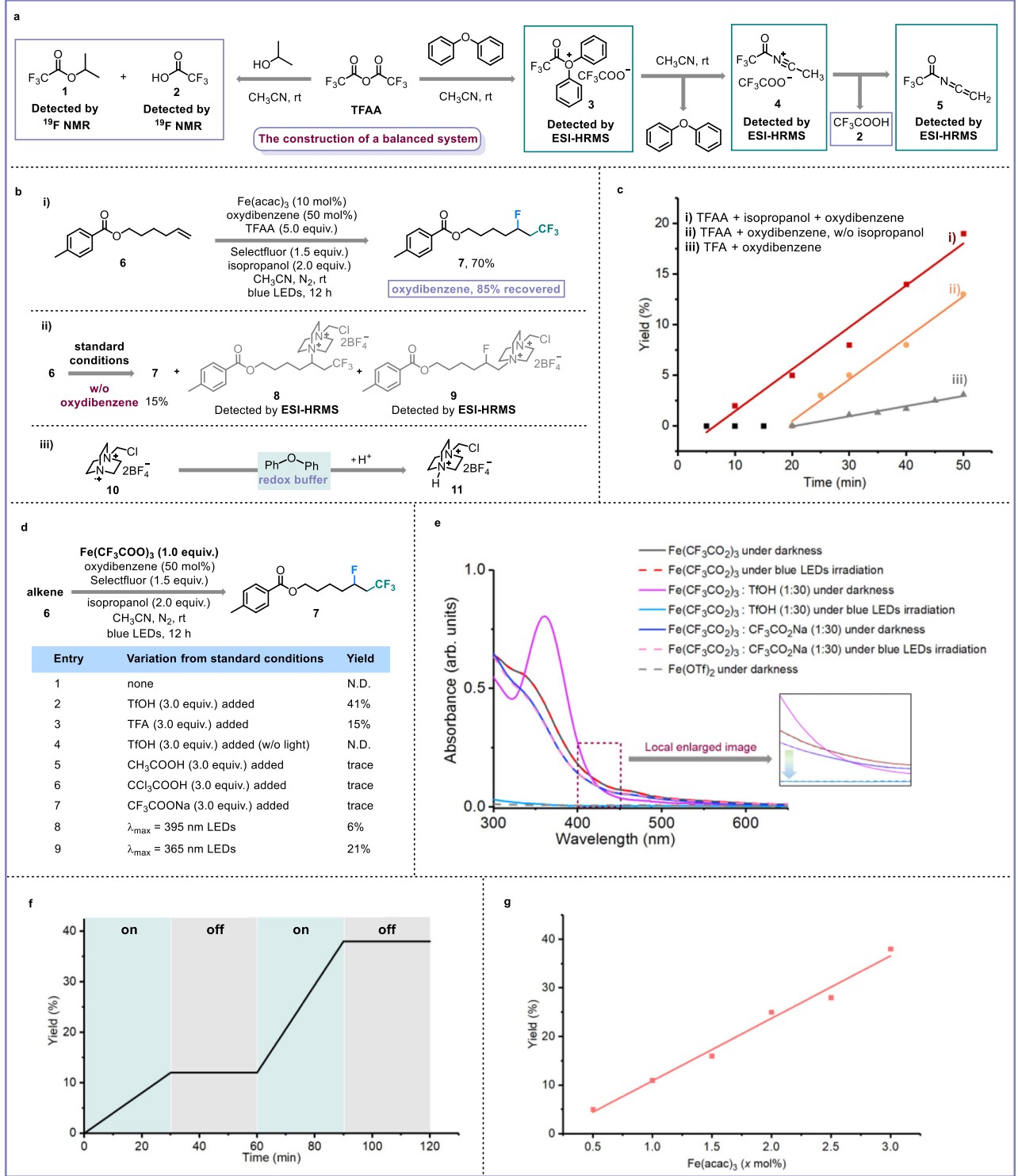

**Fig. 2 | Design and identification of Brønsted acid-unlocked iron ligand-to-metal charge transfer. a** In situ generating Brønsted acid and $CF_3COO^-$. **b** Oxydibenzene as redox buffer. **i Standard conditions**: alkene (**6**, 0.2 mmol), $(CF_3CO)_2O$ (1.0 mmol), Selectfluor (0.3 mmol), Fe(acac)$_3$ (0.02 mmol), oxydibenzene (0.1 mmol), isopropanol (0.4 mmol), and $CH_3CN$ (0.1 M), $N_2$, blue LEDs, 12-h reaction time; (**ii**) without oxydibenzene under standard conditions; (**iii**) with

oxydibenzene as redox buffer. **c** The advantage of balanced system. (**i**) standard conditions; (**ii**) without isopropanol under standard conditions; (**iii**) with TFA instead of TFAA and isopropanol in standard conditions. **d** Stoichiometric experiments of Fe(III)-intermediate species. TfOH trifluoromethanesulfonic acid, N.D. not detected. **e** UV–Vis experiments. **f** Light on/off experiments. **g** Kinetic studies of Fe(acac)$_3$.

and TFA to generate Fe(CF$_3$COO)$_3$ in situ and to create an acidic atmosphere also gave similar UV−Vis results (Supplementary Fig. 27). We also found that the amount of Brønsted acid determined the yield of fluorotrifluoromethylation−three equivalents relative to the Fe(CF$_3$COO)$_3$ loading should be optimal for CF$_3$ radical production in

the stoichiometric experiments (Supplementary Fig. 30). These results certainly reveal that a strong Brønsted acid is indeed capable of unlocking the challenging LMCT of Fe(CF$_3$COO)$_3$ for CF$_3$ radical generation and Fe(CF$_3$COO)$_3$ combined with H$^+$ under our standard conditions is a potential LMCT species. The requirement of continuous

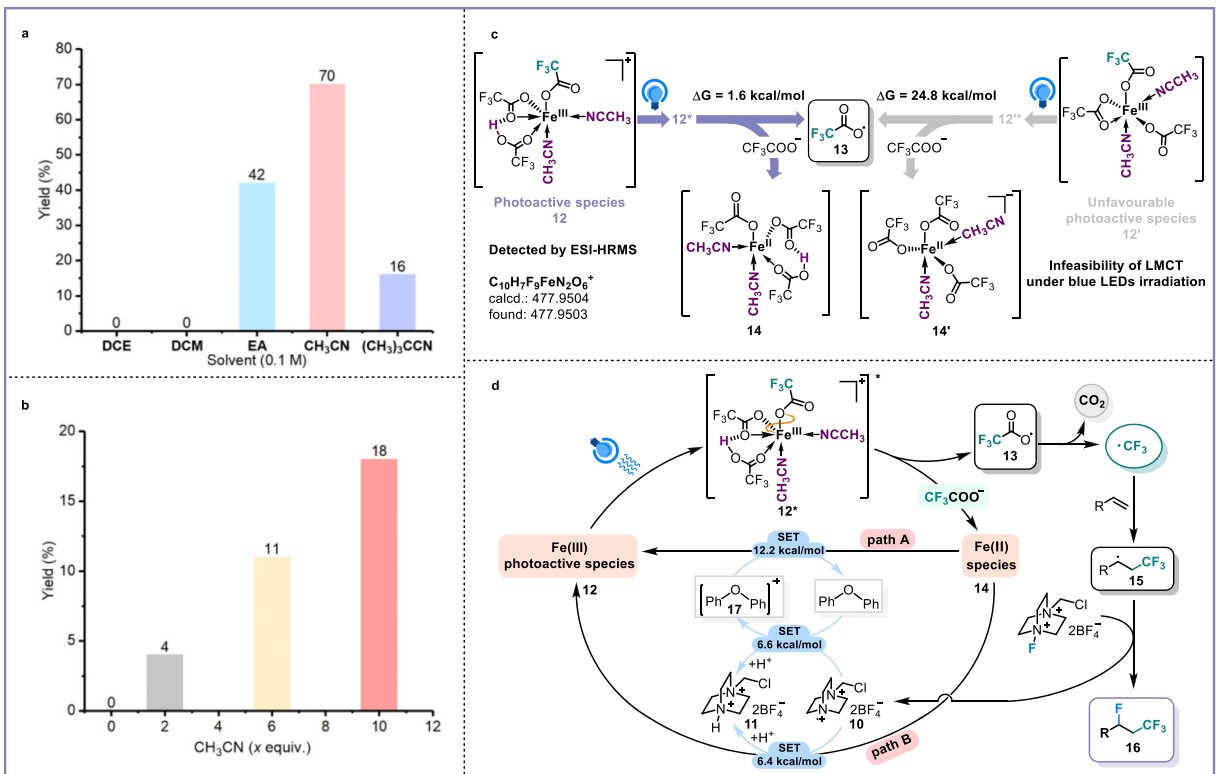

**Fig. 3 | Identification of iron-based light-harvesting species. a** Comparison of several different solvents. **b** Identification of CH₃CN as ligands. **c** Density Functional Theory (DFT) calculations. **d** Possible mechanism.

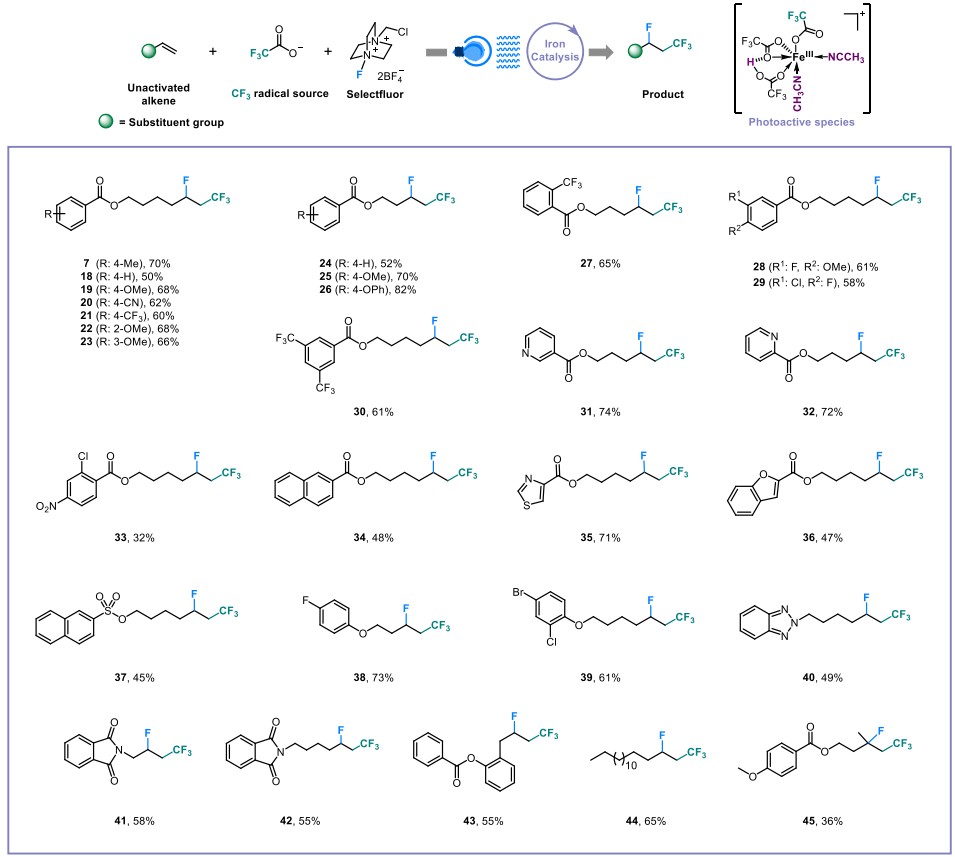

**Fig. 4 | Scope of unactivated alkenes.** ᵃGeneral reaction conditions: alkene (0.2 mmol), (CF₃CO)₂O (1.0 mmol), Selectfluor (0.3 mmol), Fe(acac)₃ (0.02 mmol), oxydibenzene (0.1 mmol), isopropanol (0.4 mmol) and CH₃CN (0.1 M), N₂, blue LEDs, 12-h reaction time.

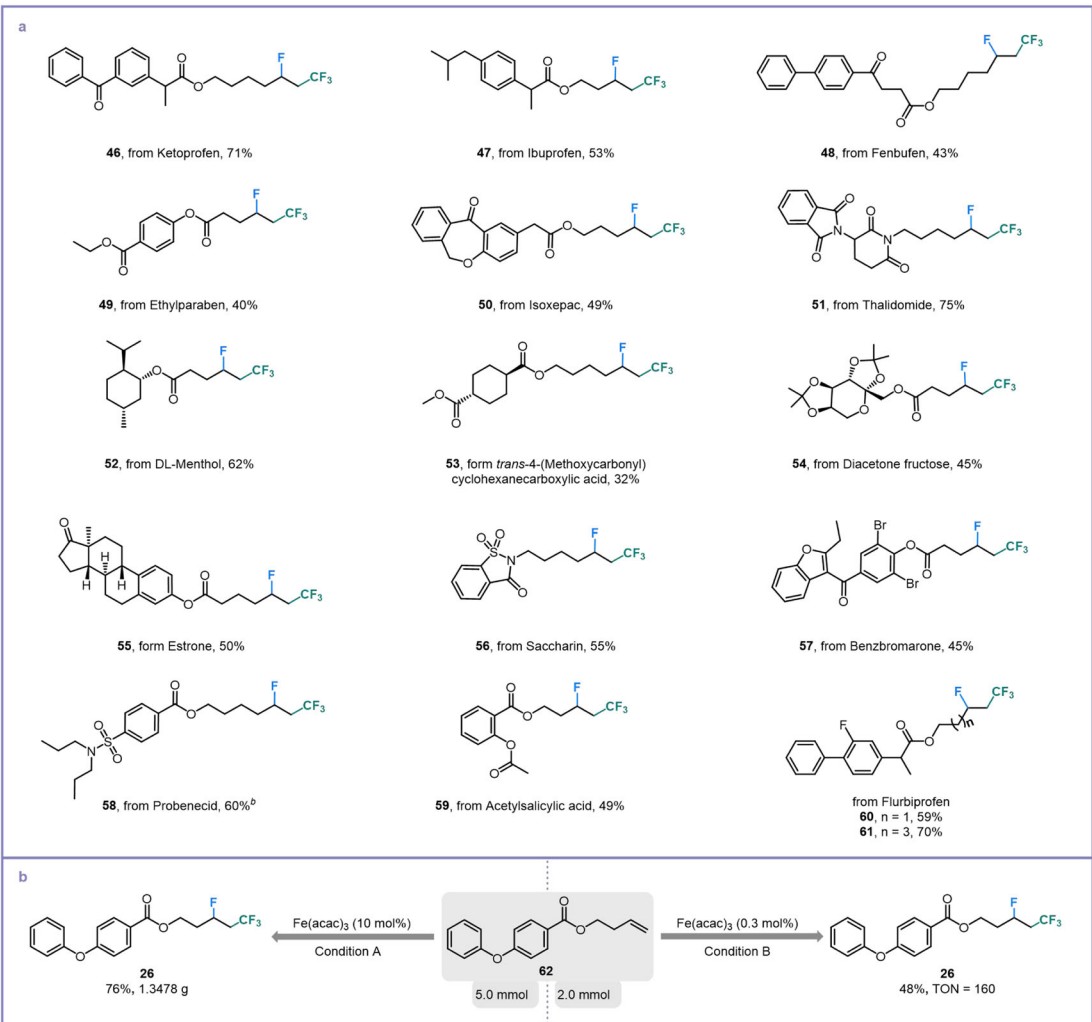

**Fig. 5 | Late-stage functionalization and scale-up experiments. a** Modification of pharmaceutical derivatives. [a]General reaction conditions: alkene (0.2 mmol), (CF₃CO)₂O (1.0 mmol), Selectfluor (0.3 mmol), Fe(acac)₃ (0.02 mmol), oxydibenzene (0.1 mmol), isopropanol (0.4 mmol) and CH₃CN (0.1 M), N₂, blue LEDs, 12-h reaction time. [b]Blue LEDs, 24-h reaction time. **b** Gram-scale and TON experiments. Condition A: alkene (**62**, 5.0 mmol), (CF₃CO)₂O (25.0 mmol), Selectfluor (7.5 mmol), Fe(acac)₃ (0.5 mmol), oxydibenzene (2.5 mmol), isopropanol (10.0 mmol) and CH₃CN (50.0 mL), N₂, blue LEDs, 72-h reaction time. Condition B: alkene (**62**, 2.0 mmol), (CF₃CO)₂O (10.0 mmol), Selectfluor (3.0 mmol), Fe(acac)₃ (0.006 mmol), oxydibenzene (1.0 mmol), isopropanol (4.0 mmol) and CH₃CN (20.0 mL), N₂, blue LEDs, 144-h reaction time. TON turnover number.

irradiation was confirmed by the light on/off experiments over time (Fig. 2f). Importantly, the kinetic curves of Fe(acac)₃ and TFAA show that the initial reaction rate would improve when increasing the loading of the iron catalyst and C$_n$X$_m$COO⁻ source (Fig. 2g, Supplementary Figs. 31 and 32).

Additionally, during the solvent investigation, we observed that only when the solvents possess coordination ability like CH₃CN, EA, and (CH₃)₃CCN, can desired fluorotrifluoromethylation product be obtained. In contrast, dichloromethane (DCM) and 1,2-dichloroethane (DCE) as solvents failed in generating CF₃ radicals (Fig. 3a). It inspired us to consider whether CH₃CN was involved in the assembly of iron and C$_n$X$_m$COO⁻-based light-harvesting species (Fig. 3a, b, Supplementary Table 3, Supplementary Fig. 33). To quickly determine the real structure of iron and C$_n$X$_m$COO⁻-based light-harvesting species, we performed density functional theory (DFT) calculations (Supplementary Figs. 34–37). As shown in Fig. 3c, the **12** that could be regarded as the combination of Fe(CF₃COO)₃ with H⁺ and two molecules of CH₃CN through the hydrogen bond and coordination effect, respectively, indeed behaves the efficient LMCT between iron and monodentate-coordinated CF₃COO⁻ under blue light irradiation. The presence of **12** was further identified by ESI-HRMS experiments

(Supplementary Fig. 24). Without the hydrogen bond effect of the Brønsted acid, the release of CF₃COO radical **13** from excited **12′** became thermodynamically unfavorable (Supplementary Figs. 38–42). Moreover, in the presence of intramolecular hydrogen bonding, the LUMO orbital energy of iron significantly decreased by 0.86 eV, which is beneficial to the desired LMCT process (Supplementary Fig. 44). Considering that the possibility of F⁻ existing in the reaction system and based on the structure of **12**, the replacement of one of hydrogen bond-binding CF₃COO⁻ groups with F⁻ to serve as an alternative but unnecessary assembly of iron-based light-harvesting species could not be excluded (see Supplementary Figs. 46–49 for detailed discussions).

The detailed mechanism cycle of this protocol was illustrated in Fig. 3d (Supplementary Figs. 52 and 53). In the presence of Brønsted acid, CF₃COO⁻ and acetonitrile, Brønsted acid-unlocked iron LMCT of proposed light-harvesting species **12** would occur under blue light irradiation, delivering CF₃COO radical **13** and Fe(II) intermediate **14**. After the release of CO₂ gas from radical intermediate **13** (Supplementary Fig. 11), the desired CF₃ radical was produced, which was further trapped by the alkene to form radical adduct **15**. Radical fluorination between **15** and Selectfluor provided the desired

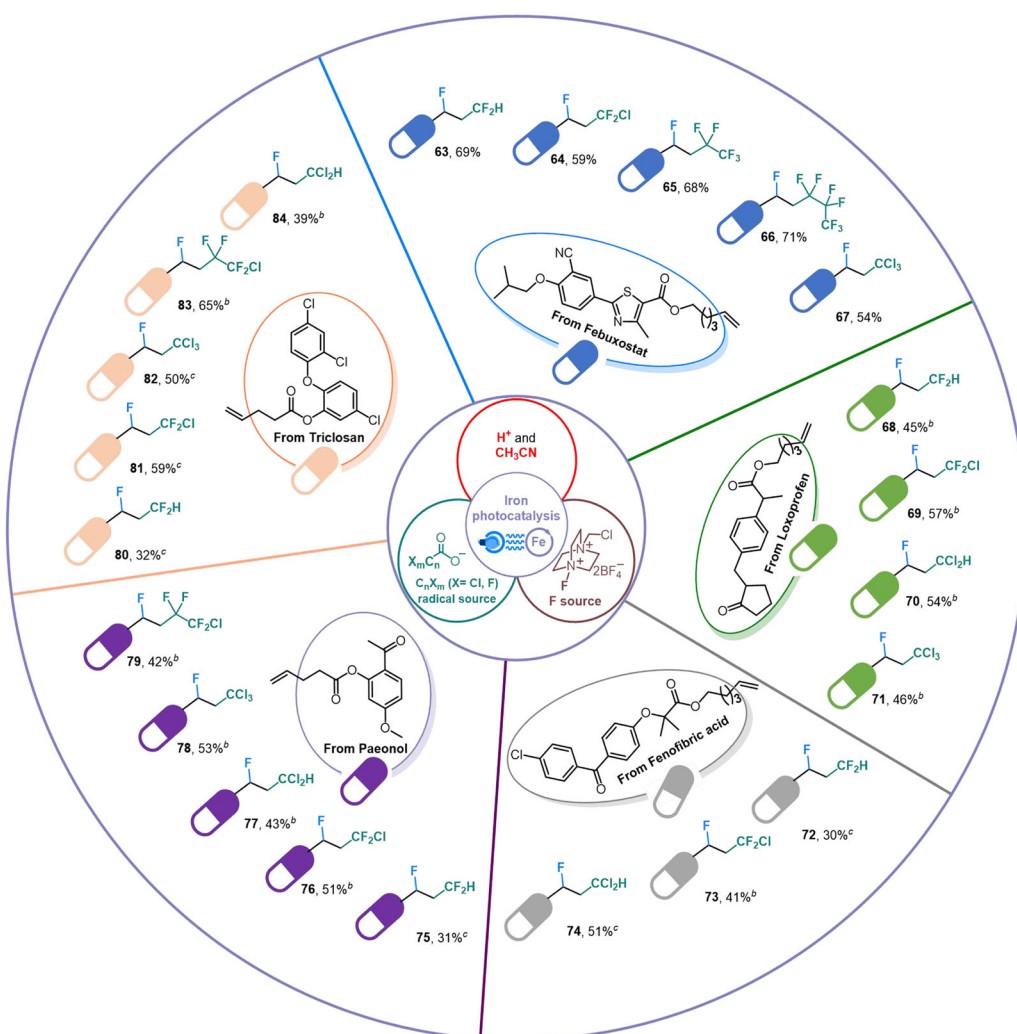

**Fig. 6 | Synthetic applications.** [a]General reaction conditions: alkene (0.2 mmol), $(C_nX_mCO)_2O$ (1.0 mmol), Selectfluor (0.3 mmol), Fe(acac)$_3$ (0.02 mmol), oxydibenzene (0.1 mmol), isopropanol (0.4 mmol) and CH$_3$CN (0.1 M), N$_2$, blue LEDs, 12- h reaction time. [b]Blue LEDs, 24-h reaction time. [c]$(C_nX_mCO)_2O$ (1.4 mmol) instead of $(C_nX_mCO)_2O$ (1.0 mmol), blue LEDs, 36-h reaction time.

fluorotrifluoromethylation product **16**. The generated *N* radical cation **10** required the regulation by the redox buffer oxydibenzene to avoid the yielding of relevant C-N bonds (Supplementary Figs. 16, 21, and 22). The radical cations **17** and **10** should be responsible for recycling iron(III) through the SET steps, which was also rationalized by the DFT calculations (Supplementary Figs. 50 and 51).

## Scope of substrates

Having established the system of Brønsted acid-unlocked iron LMCT, various valuable alkyl fluorides were produced via our method (Fig. 4). Numerous aliphatic terminal olefins containing diverse benzoate groups were well tolerated to deliver the fluorotrifluoromethylation products with satisfying yields and single chemoselectivity (**7** and **18–36**), even though the benzoate might be susceptible to reaction in the presence of a strong Brønsted acid. Besides, a sulfonate-modified alkene was also competent (**37**). Notably, the strong oxidizability of Selectfluor under our system did not hinder the transformation of those alkenes connecting electron-rich anisoles for the construction of desired products (**38** and **39**). We also assembled high-value triazole and phthalimide to alkenes, which provided moderate fluorotrifluoromethylation yields (**40–42**). The allylbenzene derivative and 1-hexadecene were also feasible (**43** and **44**). Additionally, α,α-disubstituted olefin was successfully functionalized to enrich the diversity of alkyl fluorides (**45**).

## Late-stage functionalization

To further showcase the excellent synthetic compatibility of this protocol, a wide variety of complex olefins derived from pharmaceutical molecules were subjected to this Brønsted acid-unlocked iron LMCT photocatalysis condition (Fig. 5a). The indispensable pain-relief drugs such as ketoprofen, ibuprofen, and fenbufen were installed with alkyl fluorides in moderate to good yields (**46–48**). Additionally, derivatives of the germicide ethyl-paraben and anti-inflammatory agent isoxepac could also be tolerated smoothly (**49** and **50**). It is worth noting that the fluorotrifluoromethylation of thalidomide's derivate was capable of affording 75% yield (**51**), in which thalidomide is an effective cure for the erythema nodosum leprosum. The introduction of an alkenyl group into DL-menthol (a traditional Chinese medicine for detoxification) produced a satisfying substrate for fluorotrifluoromethylation (**52**). We also extended this protocol to the transformation of the key medicine intermediates *trans*-4-(methoxycarbonyl)cyclohexanecarboxylic acid and diacetone fructose to increase the opportunity for further modifications (**53** and **54**). Alkenes with estrone and saccharin scaffolds were both converted into their corresponding products with moderate yields (**55** and **56**). Notably, benzbromarone and probenecid, drugs for the treatment of gout, contain olefin derivatives that were suitable substrates for late-stage fluorotrifluoromethylation (**57** and **58**). Additionally, antipyretic analgesics such as acetylsalicylic acid and flurbiprofen could provide good yields (**59–61**). The success

of the gram-scale and TON experiments (TON: 160) showed the application potential of this intriguing synthesis (Fig. 5b).

In addition to fluorotrifluoromethylation of alkenes, our platform of a Brønsted acid-unlocked iron LMCT could also broaden the divergent synthesis and solve the challenging radical transformations including fluoro-difluoromethylation (**63, 68, 72, 75,** and **80**), fluoro-chlorodifluoromethylation (**64, 69, 73, 76,** and **81**), fluoro-trichloromethylation (**67, 71, 78,** and **82**), fluoro-dichloromethylation (**70, 74, 77,** and **84**), fluoro-pentafluoroethylation (**65**), fluoro-chlorotetrafluoroethylation (**79** and **83**), and fluoro-heptafluoropropylation (**66**) of drug molecules (Fig. 6).

## Discussion

In conclusion, a practical Brønsted acid-unlocked iron LMCT protocol for the activation of various inert halogen-containing carboxylates to $C_nX_m$ radicals was disclosed through detailed mechanistic studies, which was applied to fascinating fluoro-polyhaloalkylation of non-activated alkenes. Hydrogen bond effect of the Brønsted acid and the coordination of the $CH_3CN$ solvent are highly important to ensure the effective assembly of iron and $C_nX_mCOO^-$-based light-harvesting species. Further studies on more challenging inert compound activation via Brønsted acid-unlocked 3d-metal LMCT are currently in progress in our laboratory.

## Methods

### General procedure for fluoro-polyhaloalkylation of alkenes

A 25 mL Schlenk flask equipped with a magnetic bar was charged with $Fe(acac)_3$ (7.1 mg, 0.02 mmol) and Selectfluor (106.3 mg, 0.3 mmol). The flask was evacuated and refilled with $N_2$ for three times. The vessel was then charged with extra dry $CH_3CN$ (2.0 mL), alkene (0.2 mmol), $(C_nX_mCO)_2O$ (X = F or Cl) (1.0 or 1.4 mmol), oxydibenzene (17.0 mg, 0.1 mmol) and isopropanol (24.0 mg, 0.4 mmol). The reaction mixture was stirred under nitrogen atmosphere and irradiated by blue LEDs for 12 or 24 or 36 h. After completion of the reaction, $CO_2$ was detected by TCD-GC. Then the reaction system was quenched by triethylamine, diluted with EtOAc. After concentrated under vacuum, the resulting residue was purified by silica gel flash column chromatography to give the products.

## Data availability

The authors declare that the data relating to the characterization of materials and products, general methods, optimization studies, experimental procedures, mechanistic studies, HRMS data and NMR spectra, computational studies are available within the article and its Supplementary Information as well as supplementary data. All data are available from the corresponding author upon request. Source data are provided with this paper.

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

## Acknowledgements

This project was supported by the National Natural Science Foundation of China, Grant Nos. 22101265 (L.N.), 21903071 (S.L.) and 21822303 (Y.L.);

China Postdoctoral Science Foundation, Grant Nos. 2022M712866 (L.N.), 2023M733212 (S.J.); Joint Fund of Key Technologies Research & Development Program of Henan Province, Grant No. 222301420006 (Y.L.); Promotion Projects for Key Research & Development in Henan Province, Grant No. 222102310042 (L.N.); the Ministry of Science and Technology of the People's Republic of China (Y.L.).

## Author contributions

X.J. and L.N. conceived the work. X.J. and Y.H. designed the experiments and analyzed the data. X.J., Y.H., K.J., J.H., J.Z., S.J., and J.S. performed the synthetic experiments. Y.L. and S.L. contributed to the DFT calculations. X.J. and L.N. described original manuscript and all authors revised.

## Competing interests

The authors declare no competing interests.
