## [Peer Review File · Nature Communications]

Iron photocatalysis via Brønsted acid-unlocked ligand-to-metal charge transferEditorial Note: Parts of this Peer Review File have been redacted as indicated to remove third-party material where no permission to publish could be obtained.

REVIEWER COMMENTS

Reviewer #1 (Remarks to the Author):

Lan, Niu, and coworkers have demonstrated a photo-promoted fluoro-polyhaloalkylation of alkenes via ligand-to-metal charge transfer (LMCT) of iron (III) haloalkylcarboxylate complexes. Subsequent decarboxylation of the resulting haloalkylcarboxyl radicals allowed access to the usually inaccessible haloalkyl radicals. Alkene trapped the generated haloalkyl radical to deliver a carbon-centered radical, which interacted with NFSI to afford the final product. The authors showed that Brønsted acids had a unique effect in turning on reactivity. The method featured a broad alkenes scope of alkyl olefins and bioactive molecules with fluoro-trifluoromethylation. Additionally, other haloalkylations were shown on three late-stage drug molecules. With the interesting reactivity and its application to complex molecules, I recommend this paper to be published in Nature Communications after the following revisions.

1. While the authors demonstrated that in-situ generated Brønsted acid were effective in promoting the reaction, all the controlled experiments were run in presence of oxydibenzene. It is unclear what oxydibenzene's role is in the reaction. Can the authors run control experiments in absence of oxydibenzene, TFAA, and isopropanol while adding common Brønsted acids, such as TfOH?
2. The control experiment in absence of oxydibenzene (Figure 2Cb) showed 15% product and the major byproduct 7. What is the mass balance of this reaction? If NFSI is used instead, would the NSI-adduct be detected?
3. For the UV-visible experiment with $\text{Fe}(\text{CF}_3\text{COO})_3$ and CF_3COONa (Figure 2E), will reactivity be turned on if the authors add TfOH to the reaction mixture? will the UV-vis profile change?
4. The reaction mechanism is unclear from the presented control experiments. Iron (II) was reported to activates NFSI previously (J. Am. Chem. Soc. 2016, 138, 12771–12774). Did the authors observe similar reactivity with Selectfluor? Can the authors perform stoichiometric experiment with $\text{Fe}(\text{CF}_3\text{COO})_3 + \text{TfOH}$ in the dark then irradiate to generate Fe(II) then add Selectfluor to monitor the reaction via UV-vis?
5. For the substrate scope, can the authors comment on other identified byproducts?

Reviewer #2 (Remarks to the Author):

See attached document.

Reviewer #3 (Remarks to the Author):

The manuscript by Jiang et. al. describes the photocatalytic activation of trifluoroacetate and other haloalkylcarboxylates that is attributed to dissociative LMCT excitation of in situ generated Fe(III) complexes in the presence of Brønsted acids. In combination with a fluorine radical source, fluoro-polyhaloalkylated products can be obtained from non-activated alkenes including pharmaceutically relevant structures.

Photocatalytic reactions exploiting dissociative LMCT for organic synthesis are attracting significant interest and I can imagine that the present work makes a valuable addition to the increasing repertoire of synthetic protocols. Conceptionally, the present work is however very similar to other photocatalytic protocols based on the generation of various radicals via dissociative LMCT. For the specific challenge of photochemical CF₃COOH activation, alternative approaches have been previously reported (Stephenson, Nat Commun 6, 7919 (2015)) and a recent preprint (DOI: 10.26434/chemrxiv-2023-j1bwd) also describes a case of CF₃COOH activation that is supposed to proceed via LMCT excitation of in situ generated Fe(III) complexes.

In this work, the authors emphasize very much the importance of Brønsted acid for the photoreaction. In this regard, a major shortcoming is the lacking characterization of the photoactive species and the role of the Brønsted acid for its reactivity. The available data is limited to UV-vis spectra (Fig. 2E) and does not provide much insight. Consequently, the characterization of the photoactive species as "Fe(CF₃COOO)₃ combined with acids" that are "unlocking" the MLCT reaction remains much too vague in my view. It also remains unclear to me why haloalkylcarboxylates and their conjugate acid are not added directly but have to be provided by the "construction of a balanced system" (Fig. 2A, B).

In summary, I can imagine that the protocol reported in this manuscript might be interesting to a more specialized audience. I am however convinced that more insight into the nature of the photoactive species and the role of the Brønsted acid would be necessary to make this work more suitable for publication in Nature Communications.

The current manuscript by X. Jiang et al. discloses a useful method for the selective fluorotrifluoromethylation of terminal alkenes making use of Fe photocatalysts, TFAA (source of trifluoromethylating radical), Selectfluor (source of fluorinating radical) and blue light. The scope of this transformation is very impressive being: 1) compatible with a good number of functionalities; 2) efficient in late stage functionalization; and 3) extensible to a broad family of polyhalogenated groups (CF_2H , CF_2Cl , CCl_2H , CCl_3 , $\text{CF}_2\text{CF}_2\text{Cl}$, C_2F_5 , C_3F_7 ,...). Thus, just in terms of synthetic utility and reaction discovery, this work deserves attention and dissemination in an important forum like this one. This said, this reviewer has doubts about the proposed mechanism of the reaction, and disagrees at this point with the main claim of this work, i.e. the absolute need of a Brønsted acid ($\text{CF}_3\text{CO}_2\text{H}$) to induce the CF_3 radical generation and CO_2 extrusion from the trifluoroacetate group via “proton mediated” MLCT photocatalysis. Also, the citation is incorrect as relevant articles dealing with the in situ production of easy-to-reduce $\text{CF}_3\text{CO}_2\text{-LG}$ fragments (LG = appropriate leaving group) are missing and must be cited. See for instance: C. R. J. Stephenson and coworkers, *Nat. Commun.* **2015**, *6*, 7919, doi: [10.1038/ncomms8919](https://doi.org/10.1038/ncomms8919) & *Chem* **2016**, *1*, 456, doi: [10.1016/j.chempr.2016.08.002](https://doi.org/10.1016/j.chempr.2016.08.002); J. Jin and co-workers, *Cell Reports Physical Science* **2020**, *1*, 100141, doi: [10.1016/j.xcrp.2020.100141](https://doi.org/10.1016/j.xcrp.2020.100141); W. Su and coworkers, *Chem Catalysis* **2022**, *2*, 1793, doi: [10.1016/j.checat.2022.05.018](https://doi.org/10.1016/j.checat.2022.05.018). In sharp contrast, references 62-64 are cited in current manuscript to support the Brønsted acid mediated photocatalysis concept, but those references are irrelevant here as they do not deal with the participation of metallic photocatalysts. Taken together, I am unable to support the acceptance of this manuscript in *Nature Communications* in its current form, although it could be publishable after major revisions to provide a more profound mechanistic study.

Main points for the revision:

1.- Additional support must be provided regarding the requirement of $\text{CF}_3\text{CO}_2\text{H}$ (formed *in situ* from TFAA and *i*PrOH) to unlock the iron LMCT process under light irradiation to release CF_3 radicals. Recently, a preprint by F. Julia-Hernandez has appeared reporting trifluoromethylation of arenes via MLCT process using NaO_2CCF_3 reagent photocatalyzed by $(\text{N}^{\wedge}\text{N})\text{Fe}(\text{O}_2\text{CCF}_3)_3$ under blue light illumination (see F. Julia-Hernandez and coworkers, *ChemRxiv* **2023**, doi: [10.26434/chemrxiv-2023-j1bwd](https://doi.org/10.26434/chemrxiv-2023-j1bwd)). This work shows how $\text{Fe}^{\text{III}}\text{-O}_2\text{CCF}_3$ bond breaks under blue light to release CF_3 radicals and CO_2 thus allowing efficient trifluoromethylation. Stoichiometric investigations proved the higher activity of $(\text{N}^{\wedge}\text{N})\text{Fe}(\text{O}_2\text{CCF}_3)_3$ (88% combined yield) compared to the one of $\text{Fe}(\text{O}_2\text{CCF}_3)_3$ (22% yield) and clearly illustrates the crucial role of the ancillary ligand in this transformation. The current manuscript by X. Jiang et al. claims the opposite, and suggests the participation of $\text{CF}_3\text{CO}_2\text{H}$ in the LMCT process. However, in the current work, the true nature of the iron catalyst remains unknown. To this regard, the effective cocktail reported here brings along with it several compounds (acetylacetone, TFAA, *i*PrOH, CH_3CN) that may act as ancillary ligands thus tuning the activity of the iron photocatalyst. The $\text{CF}_3\text{CO}_2\text{H}$ may just participate as a source of protons to quench the nitrogen radical cation formed after fluorination, or alternative, to avoid the coordination of the basic ligands mentioned above. The authors carried out varied stoichiometric experiments (Figure 2D and 2F) to support the requirement of $\text{CF}_3\text{CO}_2\text{H}$ in the MLCT process, but the reaction conditions considerably differs from the ones used in the catalytic experiments, as large excess of *i*PrOH is used in Figures 2D and 2F, while *i*PrOH is quenched by TFAA under catalytic conditions. To this regard, does the reaction work when replacing *i*PrOH by $[\text{nBu}_4\text{N}][\text{iPrO}]$ salt (i.e. without $\text{CF}_3\text{CO}_2\text{H}$ formation)? Shortly, the true role of in situ formed $\text{CF}_3\text{CO}_2\text{H}$ remains unknown and needs clarification prior to definite publication of the current work.

2.- Selectfluor is strong oxidizing agent. Did the authors prove that the Fe(II)/Fe(III) redox scenario is compatible with the presence of Selectfluor? It's hard to believe that the iron photocatalyst does not

react with the large excess of Selectfluor giving rise to Fe(III)-F or Fe(IV)-F species. This reaction should occur instantaneously even in absence of light. Fe(III)-F and Fe(IV)-F complexes have been prepared previously by using hydrogen peroxide (Alexander B. Sorokin and coworkers, *J. Am. Chem. Soc.* **2014**, *136*, 11321) or PhI(F)₂ (A. R. McDonald and co-workers, *JACS Au* **2023**, *3*, 919 & *Angew. Chem. Int. Ed.* **2021**, *60*, 26281).

3.- In page 4, lines 95-99, the authors state that “*the possible trifluoromethylative amination of alkene as the major transformation was detected by ESI (Fig. S20)*” and that the oxydibenzene serves “*to quench the N-radical cation that is yielded from the radical fluorination step (Fig. 2C-b)*”. This proposal makes sense, however, if the N-radical cation comes from the radical fluorination step, the C-N coupled product can be formed at maximum in 1:1 ratio vs the desired fluorotrifluoromethylated product. How do the authors argue the observation of the C-N coupled product as the main compound in absence of oxydibenzene?

4.- The manuscript should be written in a more pedagogical manner to help readers to identify the major challenge warranting urgent publication in a major journal such as *Nature Communications*.

Other minor points:

5.- In page 5, lines 138-143, the terms “benzoyl” and “sulfonyl” are confusing. It would be better to use the terms “benzoate” and “sulfonate”.

6.- In page 6, lines 147-148, the authors refer to “*α-methyl and internal olefins ... that yields the diverse alkyl fluorides 39-41*”. However, in the corresponding Figure 3, only one α-methyl olefin is functionalized (**41**), and no example of internal olefin is shown in such Figure. The main text must be corrected accordingly.

7.- In Figure 5, reaction conditions for the functionalized products **58-62** derived from Febuxostat are missing.

Response to the comments from Reviewer #1.

Lan, Niu, and coworkers have demonstrated a photo-promoted fluoro-polyhaloalkylation of alkenes via ligand-to-metal charge transfer (LMCT) of iron (III) haloalkylcarboxylate complexes. Subsequent decarboxylation of the resulting haloalkylcarboxyl radicals allowed access to the usually inaccessible haloalkyl radicals. Alkene trapped the generated haloalkyl radical to deliver a carbon-centered radical, which interacted with NFSI to afford the final product. The authors showed that Brønsted acids had a unique effect in turning on reactivity. The method featured a broad alkenes scope of alkyl olefins and bioactive molecules with fluoro-trifluoromethylation. Additionally, other haloalkylations were shown on three late-stage drug molecules. With the interesting reactivity and its application to complex molecules, I recommend this paper to be published in Nature Communications after the following revisions.

Our Response: We thank the reviewer for the positive comments on our work.

1. While the authors demonstrated that in-situ generated Brønsted acid were effective in promoting the reaction, all the controlled experiments were run in presence of oxydibenzene.

It is unclear what oxydibenzene's role is in the reaction. Can the authors run control experiments in absence of oxydibenzene, TFAA, and isopropanol while adding common Brønsted acids, such as TfOH?

Our Response: We thank the reviewer for the suggestions and comments. In the revised manuscript, we have detailedly elaborated the role of oxydibenzene in this reaction (Fig. 2B and 2C). On the one hand, oxydibenzene can react with TFAA and generate the corresponding trichloroacetates **3**, which is beneficial to provide the CF₃COO⁻ for the assembly of iron and CF₃COO⁻-based light harvesting species. On the other hand, oxydibenzene serves as the redox buffer to timely quench the electrophilic N radical cation **10** from the selectfluor. As shown in Fig. R1, in the absence of oxydibenzene, the yield of desired fluorotrifluoromethylation product is low, while the alkene was completely consumed. Through the detection of ESI-HRMS, the possibly related amination products (**8** and **9**) were generated. Notably, Heinrich also found the fact that anisole can serve as a scavenger of **10** to prevent side reactions such as the addition of **10** to the alkene (*Chem. Eur. J.* **2019**, *25*, 2786).

[Redacted]

Fig. R1 The results in the absence of oxydibenzene.

We also ran the control experiments in absence of oxydibenzene, TFAA, and isopropanol while adding TfOH as Brønsted acid. The results were summarized in Table R1. As we can see, no desired products were delivered when TfOH replaced our balanced system.

Entry	Deviation from the standard conditions	7, Yield
1	none	N.D.
2	isopropanol added	N.D.
3	isopropanol & oxydibenzene added	N.D.
4	w/o selectfluor & Fe(acac) ₃ , darkness	N.D.
5	w/o TfOH	N.D.

Table R1 Control experiments of TfOH as Brønsted acid.

2. The control experiment in absence of oxydibenzene (Figure 2Cb) showed 15% product and the major byproduct 7. What is the mass balance of this reaction? If NFSI is used instead, would the NSI-adduct be detected?

Our Response: We thank the reviewer for the suggestions. In the revised manuscript, we described the possible side-products in the absence of oxydibenzene (Fig. 2C-b). We have tried our best to determine the yields of possibly related amination products (**8** and **9**), but the obtained compounds seemed like being mixed with the **11**/selectfluor. Therefore, we resorted to the ESI-HRMS experiments for the detection of **8** and **9**, which are probably the major byproducts of this reaction.

As shown in Table R2, when NFSI was used instead of selectfluor, the yield of **44** is low and the intermolecular carboamination of alkene with N-fluorobenzenesulfonimide was observed.

The poor reactivity of NFSI for fluorotrifluoromethylation is considered to be due to its weaker electrophilicity than selectfluor. Without the regulation of redox buffer (oxydibenzene), the fluorotrifluoromethylation of alkene became worse, while the yield of sultam **86** improved. Therefore, electron-rich oxydibenzene is responsible for timely quenching the electrophilic N radical intermediate **10** or **90**.

Entry	Deviation from reaction condition	85, Recovery	44, Yield	86, NSI-adduct
1	/	57%	13%	3%
2	w/o oxydibenzene	26%	7%	25%

Table R2 The studies of NFSI as fluorine source.

3. For the UV-visible experiment with $\text{Fe}(\text{CF}_3\text{COO})_3$ and CF_3COONa (Figure 2E), will reactivity be turned on if the authors add TfOH to the reaction mixture? will the UV-vis profile change?

Our Response: We thank the reviewer for the suggestions. As you can see (Scheme R1), the addition of TfOH indeed unlocks the reactivity of fluorotrifluoromethylation when utilizing stoichiometric $\text{Fe}(\text{CF}_3\text{COO})_3$ and CF_3COONa as CF_3 source. The UV-Vis studies also evidenced this fact. With the addition of TfOH and the irradiation of blue light, the signal of trivalent iron disappeared, indicating that the acid indeed promotes the desired LMCT.

Scheme R1 The studies of Brønsted acid.

4. The reaction mechanism is unclear from the presented control experiments. Iron (II) was reported to activate NFSI previously (J. Am. Chem. Soc. 2016, 138, 12771–12774). Did the authors observe similar reactivity with Selectfluor?

Our Response: We thank the reviewer for the suggestions and comments. According to the detailed mechanistic studies in the revised manuscript (Fig. 3B and 3C), we can conclude that the real iron-based light-harvesting species under blue light irradiation may derive from the in situ-assembly of Fe^{3+} , CF_3COO^- , H^+ and solvent acetonitrile. Due to the fact that chlorotrifluoromethylation of alkene can also be achieved when selectfluor was replaced into CCl_3CN under the same catalytic condition (Fig. R2), selectfluor thus is not involved in the generation step of CF_3 radical, which acts as halogen atom transfer (XAT) reagent like CCl_3CN .

Fig. R2 CCl3CN as the [Cl] source.

Fig. R3 The NMR spectra of **91**.

Can the authors perform stoichiometric experiment with $\text{Fe}(\text{CF}_3\text{COO})_3 + \text{TfOH}$ in the dark then irradiate to generate $\text{Fe}(\text{II})$ then add Selectfluor to monitor the reaction via UV-vis?

Our Response: We thank the reviewer for the suggestion. As we can see (Fig. R4), Brønsted acid TfOH can indeed induce the photolysis of $\text{Fe}(\text{CF}_3\text{COO})_3$ to $\text{Fe}(\text{II})$ under blue LEDs irradiation, while selectfluor can not effectively promote the regeneration of $\text{Fe}(\text{III})$. Only when the oxydibenzene and trichloroacetate are in the presence of $[\text{Fe}(\text{II})]$ and selectfluor, can the signal of $[\text{Fe}(\text{III})]$ be observed. This result further supports our conclusion ($[\text{Fe}(\text{II})]/[\text{Fe}(\text{III})]$ redox cycle), because the addition of electron-rich oxydibenzene may induce the electrophilic fluorination of selectfluor and thus generate the N radical cation **10** (*J. Chem. Soc., Perkin Trans. 2*, **2002**, 953–957; *CCS Chem.* **2020**, 2, 566), which then oxidizes the $\text{Fe}(\text{II})$ to $\text{Fe}(\text{III})$. Because the CF_3COO^- is consumed during the iron LMCT process, the replenishment of CF_3COO^- is beneficial to the regeneration of $\text{Fe}(\text{CF}_3\text{COO})_3$.

A) The UV-Vis monitorization of the oxidation of $\text{Fe}(\text{II})$ to $\text{Fe}(\text{III})$

- (1) $\text{Fe}(\text{CF}_3\text{CO}_2)_3$ [0.25 mM].
- (2) $\text{Fe}(\text{CF}_3\text{CO}_2)_3$ + TfOH (1 : 30).
- (3) $\text{Fe}(\text{CF}_3\text{CO}_2)_3$ + TfOH (1 : 30) under blue LEDs irradiation.
- (4) $\text{Fe}(\text{CF}_3\text{CO}_2)_3$ + TfOH under blue LEDs irradiation, then, selectfluor (1 : 30 : 15) added.
- (5) $\text{Fe}(\text{CF}_3\text{CO}_2)_3$ + TfOH under blue LEDs irradiation, then, selectfluor + oxydibenzene (1 : 30 : 15 : 5) added.
- (6) $\text{Fe}(\text{CF}_3\text{CO}_2)_3$ + TfOH under blue LEDs irradiation, then, $\text{CF}_3\text{CO}_2\text{Na}$ + selectfluor + oxydibenzene (1 : 30 : 10 : 15 : 5) added.

B) The UV-Vis monitorization of the oxidation of $\text{Fe}(\text{III})$ to $\text{Fe}(\text{II})$

Fig. R4 Brønsted acid inducing the photolysis of $\text{Fe}(\text{CF}_3\text{COO})_3$.

5. For the substrate scope, can the authors comment on other identified byproducts?

Our Response: We thank the reviewer for the comments. During the investigation of substrate scope, we could obviously observe the C(sp²)-H trifluoromethylation of **19** owing to the presence of electron-rich arene moiety (Fig. R5).

Fig. R5 Other identified byproduct.

Fig. R6 The NMR spectra of 94.

Response to the comments from Reviewer #2.

The current manuscript by X. Jiang et al. discloses a useful method for the selective fluorotrifluoromethylation of terminal alkenes making use of Fe photocatalysts, TFAA (source of trifluoromethylating radical), Selectfluor (source of fluorinating radical) and blue light. The scope of this transformation is very impressive being: 1) compatible with a good number of functionalities; 2) efficient in late stage functionalization; and 3) extensible to a broad family of polyhalogenated groups (CF_2H , CF_2Cl , CCl_2H , CCl_3 , $\text{CF}_2\text{CF}_2\text{Cl}$, C_2F_5 , C_3F_7 ,...). Thus, just in terms of synthetic utility and reaction discovery, this work deserves attention and dissemination in an important forum like this one.

Our Response: We thank the reviewer for the positive comments on our work.

This said, this reviewer has doubts about the proposed mechanism of the reaction, and disagrees at this point with the main claim of this work, i.e. the absolute need of a Brønsted acid ($\text{CF}_3\text{CO}_2\text{H}$) to induce the CF_3 radical generation and CO_2 extrusion from the trifluoroacetate group via “proton mediated” MLCT photocatalysis.

Our Response: We thank the reviewer for the comments. In the revised manuscript (Fig. 2F, 3A, 3B and 3C, Fig. S30, S40, S41, S43, S49 and S50), we have demonstrated the detailed mechanistic studies including DFT calculations, UV-Vis and ESI-HRMS experiments, etc. We now can conclude that the real iron-based light-harvesting species under blue light irradiation may derive from the in situ-assembly of Fe^{3+} , CF_3COO^- , H^+ and solvent acetonitrile.

Also, the citation is incorrect as relevant articles dealing with the in situ production of easy-to-reduce $\text{CF}_3\text{CO}_2\text{-LG}$ fragments (LG = appropriate leaving group) are missing and must be cited. See for instance: C. R. J. Stephenson and coworkers, Nat. Commun. 2015, 6, 7919, doi: 10.1038/ncomms8919 & Chem 2016, 1, 456, doi: 10.1016/j.chempr.2016.08.002; J. Jin and co-workers, Cell Reports Physical Science 2020, 1, 100141, doi: 10.1016/j.xcrp.2020.100141; W. Su and coworkers, Chem Catalysis 2022, 2, 1793, doi: 10.1016/j.cheecat.2022.05.018. In sharp contrast, references 62-64 are cited in current manuscript to support the Brønsted acid mediated photocatalysis concept, but those references are irrelevant here as they do not deal with the participation of metallic photocatalysts.

Our Response: We thank the reviewer for pointing out such mistakes, which we now have been corrected in the revised manuscript. All the references have been checked carefully to make sure the correctness (see references 59-62).

Taken together, I am unable to support the acceptance of this manuscript in Nature Communications in its current form, although it could be publishable after major revisions to provide a more profound mechanistic study.

Our Response: We thank the reviewer for the comments and suggestions. After the major revisions, in the revised manuscript, we indeed provided a more profound mechanistic studies to evidence and describe the real iron and $C_nX_mCOO^-$ -based light-harvesting species for the valuable C_nX_m radical production (Fig. 2F, 3A, 3B and 3C, Fig. S30, S40, S41, S43, S49 and S50). Therefore, we believe that the revised manuscript should be suitable to publish in Nature Communications.

Main points for the revision:

1.- Additional support must be provided regarding the requirement of CF_3CO_2H (formed in situ from TFAA and *i*PrOH) to unlock the iron LMCT process under light irradiation to release CF_3 radicals. Recently, a preprint by F. Julia-Hernandez has appeared reporting trifluoromethylation of arenes via MLCT process using NaO_2CCF_3 reagent photocatalyzed by $(N^{\wedge}N)Fe(O_2CCF_3)_3$ under blue light illumination (see F. Julia-Hernandez and coworkers, ChemRxiv 2023, doi: 10.26434/chemrxiv-2023-j1bwd). This work shows how $Fe^{III}-O_2CCF_3$ bond breaks under blue light to release CF_3 radicals and CO_2 thus allowing efficient trifluoromethylation. Stoichiometric investigations proved the higher activity of $(N^{\wedge}N)Fe(O_2CCF_3)_3$ (88% combined yield) compared to the one of $Fe(O_2CCF_3)_3$ (22% yield) and clearly illustrates the crucial role of the ancillary ligand in this transformation. The current

manuscript by X. Jiang et al. claims the opposite, and suggests the participation of CF_3CO_2H in the LMCT process. However, in the current work, the true nature of the iron catalyst remains unknown. To this regard, the effective cocktail reported here brings along with it several compounds (acetylacetone, TFAA, *i*PrOH, CH_3CN) that may act as ancillary ligands thus tuning the activity of the iron photocatalyst.

Our Response: We thank the reviewer for the comments and suggestions. We do appreciate and sympathize with the viewpoint that ancillary ligands should be subsistent to tune the activity of iron and $C_nX_mCOO^-$ -based light-harvesting species. In the revised manuscript (Fig. 2F, 3A, 3B and 3C, Fig. S30, S40, S41, S43, S49 and S50), we have supplemented the detailed mechanistic studies including DFT calculations, UV-Vis and ESI-HRMS experiments, etc., evidencing that the real iron-based light-harvesting species under blue light irradiation may derive from the in situ-assembly of Fe^{3+} , $C_nX_mCOO^-$, H^+ and solvent acetonitrile, in which the effect of Brønsted acid via the hydrogen-bond interaction indeed increase the efficiency of LMCT between iron center and $C_nX_mCOO^-$.

The CF_3CO_2H may just participate as a source of protons to quench the nitrogen radical cation formed after fluorination, or alternative, to avoid the coordination of the basic ligands mentioned above.

Our Response: We thank the reviewer for the comments. In the revised manuscript, we have shown that the nitrogen radical cation should be quenched by the $[\text{Fe}^{\text{II}}]$ /oxydibenzene (Fig.

3D and Fig. S47). If we replace the selectfluor into CCl_3CN , the desired chlorotrifluoromethylation of alkene could also be achieved in the Brønsted acid dominated conditions, and the acidic conditions are necessary (Table R3). According to the detailed mechanistic studies in the revised manuscript (Fig. 2F, 3A, 3B and 3C, Fig. S30, S40, S41, S43, S49 and S50), we indeed found that Brønsted acid unlocks the effective LMCT of the real iron-based light-harvesting species **12**.

Table R3 Brønsted acid unlocking the iron photocatalysis for CF_3 radical generation.

The authors carried out varied stoichiometric experiments (Figure 2D and 2F) to support the requirement of $\text{CF}_3\text{CO}_2\text{H}$ in the MLCT process, but the reaction conditions considerably differs from the ones used in the catalytic experiments, as large excess of *i*PrOH is used in Figures 2D and 2F, while *i*PrOH is quenched by TFAA under catalytic conditions. To this regard, does the reaction work when replacing *i*PrOH by $[\text{nBu}_4\text{N}][\text{iPrO}]$ salt (i.e. without $\text{CF}_3\text{CO}_2\text{H}$ formation)?

Our Response: We thank the reviewer for the comments and suggestions. By the monitoring of ^{19}F -NMR (Fig. 2A and 2D, Fig. S19 and S20), we believe that the conversion of *i*PrOH and TFAA to TFA and ester **1** is undoubted, which is responsible for the construction of Brønsted acid-based conditions. Directly using iPrO^- instead of *i*PrOH, no fluorotrifluoromethylation product was generated (Table R4). During the study of the stoichiometric experiments, if we remove the *i*PrOH, we could still observe the generation of desired fluorotrifluoromethylation product, but the yield is reduced (Table R5). According to the previous report (*Nat. Commun.*

2019, *10*, 467), *i*PrOH can also quench the N radical cation **10** despite with a moderate reactivity. Considering the fact that the concentration of **10** in stoichiometric experiments should be higher than catalytic conditions, therefore, *i*PrOH here may serve as the reductants to assist oxydibenzene to reduce the C-N bond formation (**8/9**) of the N radical cation and alkenes.

Entry	additive	7, Yield
1		70%
2 ^b		N.D.

^bWe use lithium isopropoxide to avoid the formation of CF₃COOH

Table R4 The necessity of isopropanol.

Entry	Deviation from the standard conditions	7, Yield
1	/	41%
2	w/o isopropanol	18%
3	w/o oxydibenzene	21%
4	w/o oxydibenzene & isopropanol	11%
5	lithium isopropoxide instead of isopropanol	N.D.

Table R5 The results of stoichiometric experiments.

Shortly, the true role of in situ formed CF₃CO₂H remains unknown and needs clarification prior to definite publication of the current work.

Our Response: To address this issue, significant mechanistic experiments have been conducted to evidence and describe the real iron and C_nX_mCOO⁻-based light-harvesting species for the valuable C_nX_m radical production in the revised manuscript (Fig. 3A, 3B and 3C, Fig. S30, S40, S41, S43, S49 and S50).

2.- Selectfluor is strong oxidizing agent. Did the authors prove that the Fe(II)/Fe(III) redox scenario is compatible with the presence of Selectfluor? It's hard to believe that the iron photocatalyst does not react with the large excess of Selectfluor giving rise to Fe(III)-F or Fe(IV)-F species. This reaction should occur instantaneously even in absence of light. Fe(III)-F and Fe(IV)-F complexes have been prepared previously by using hydrogen peroxide (Alexander B. Sorokin and coworkers, J. Am. Chem. Soc. 2014, 136, 11321) or PhI(F)₂ (A. R. McDonald and co-workers, JACS Au 2023, 3, 919 & Angew. Chem. Int. Ed. 2021, 60, 26281).

Our Response: We thank the reviewer for the comments and suggestions. In the revised manuscript, we have disclosed that the oxidation of [Fe^{II}] intermediate should be dominated

by the **10** and oxydibenzene radical cation **17** (Fig. 3D and Fig. S47). Although selectfluor is a strong oxidizing agent, we did not observe the regeneration of $[\text{Fe}^{\text{III}}]$ from the $[\text{Fe}^{\text{II}}]$ through the UV-Vis studies. Only when the oxydibenzene and trichloroacetate are in the presence of $[\text{Fe}^{\text{II}}]$ and selectfluor, can the signal of $[\text{Fe}^{\text{III}}]$ be observed (Fig. R7). This result further supports our conclusion ($[\text{Fe}^{\text{II}}]/[\text{Fe}^{\text{III}}]$ redox cycle), because the addition of electron-rich oxydibenzene may induce the electrophilic fluorination of selectfluor and thus generate the N radical cation **10** (*J. Chem. Soc., Perkin Trans. 2*, **2002**, 953–957; *CCS Chem.* **2020**, 2, 566), which then oxidizes the $\text{Fe}(\text{II})$ to $\text{Fe}(\text{III})$. Because the CF_3COO^- is consumed during the iron LMCT process, the replenishment of CF_3COO^- is beneficial to the regeneration of $\text{Fe}(\text{CF}_3\text{COO})_3$.

A) The UV-Vis monitorization of the oxidation of $\text{Fe}(\text{II})$ to $\text{Fe}(\text{III})$

- (1) $\text{Fe}(\text{CF}_3\text{CO}_2)_3$ [0.25 mM].
- (2) $\text{Fe}(\text{CF}_3\text{CO}_2)_3 + \text{TfOH}$ (1 : 30).
- (3) $\text{Fe}(\text{CF}_3\text{CO}_2)_3 + \text{TfOH}$ (1 : 30) under blue LEDs irradiation.
- (4) $\text{Fe}(\text{CF}_3\text{CO}_2)_3 + \text{TfOH}$ under blue LEDs irradiation, then, selectfluor (1 : 30 : 15) added.
- (5) $\text{Fe}(\text{CF}_3\text{CO}_2)_3 + \text{TfOH}$ under blue LEDs irradiation, then, selectfluor + oxydibenzene (1 : 30 : 15 : 5) added.
- (6) $\text{Fe}(\text{CF}_3\text{CO}_2)_3 + \text{TfOH}$ under blue LEDs irradiation, then, $\text{CF}_3\text{CO}_2\text{Na} + \text{selectfluor} + \text{oxydibenzene}$ (1 : 30 : 10 : 15 : 5) added.

B) The UV-Vis monitorization of the oxidation of $\text{Fe}(\text{III})$ to $\text{Fe}(\text{II})$

Fig. R7 Brønsted acid inducing the photolysis of $\text{Fe}(\text{CF}_3\text{COO})_3$.

As the reviewer mentioned, considering that the possibility of F^- existing in the reaction system, thus, for the alternative assembly of iron and $C_nX_mCOO^-$ -based light-harvesting species and based on the structure of **12**, replacing one of hydrogen bond-binding CF_3COO^- with fluorine to **87** could not be excluded (Fig. S31, S44, S45 and S46).

The possible catalyst species with the F^- (**87**)

Fig. R8 The calculated structure of **87**.

Fig. R9 The UV-Vis absorption predicted by the TD-DFT (**87**).

ESI-HRMS detection of **87**

Formula (87)	Exact mass	M + [H ⁺]	M + [Na ⁺]
C ₁₀ H ₇ F ₁₀ FeN ₂ O ₆	496.9494	497.9567	519.9386

a) $Fe(CF_3CO_2)_3 + TFA + TBAF$ (1 : 3 : 3) in CH_3CN , detected by ESI-HRMS,

found: 519.9398

b) $\text{Fe}(\text{CF}_3\text{CO}_2)_3 + \text{TfOH} + \text{TBAF}$ (1 : 3 : 3) in CH_3CN , detected by ESI-HRMS,
found: 497.9581

c) $\text{Fe}(\text{acac})_3 + \text{TFAA} + \text{isopropanol} + \text{TBAF}$ (1 : 5 : 2 : 3) in CH_3CN , detected
by ESI-HRMS, found: 519.9399

Fig. R10 ESI-HRMS experiments for **87**.

3.- In page 4, lines 95-99, the authors state that “the possible trifluoromethylative amination of alkene as the major transformation was detected by ESI (Fig. S20)” and that the oxydibenzene serves “to quench the N-radical cation that is yielded from the radical fluorination step (Fig. 2C-b)”. This proposal makes sense, however, if the N-radical cation comes from the radical fluorination step, the C-N coupled product can be formed at maximum in 1:1 ratio vs the desired fluorotrifluoromethylated product. How do the authors argue the observation of the C-N coupled product as the main compound in absence of oxydibenzene?

Our Response: We thank the reviewer for the comments and suggestions. After the further ESI-HRMS experiments (Fig. S22, S27 and S28), we found the C-N coupled compounds **8** and **9** possibly as the main side products. Based on the previous report, the N-radical cation **10** generated from the radical fluorination step possesses strong electrophilicity. Once the N-radical cation **10** is not timely quenched, the radical relay with alkene will produce the radical adduct **95**, which can induce another radical fluorination process to **9** and **10** (Fig. R11) (*J. Am. Chem. Soc.* **2016** *138*, 6598). This radical propagation is uncontrollable if the redox buffer oxydibenzene is absent.

Figure. R11 The propagation of N radical cation in the absence of oxydibenzene.

4.- The manuscript should be written in a more pedagogical manner to help readers to identify the major challenge warranting urgent publication in a major journal such as Nature Communications.

Our Response: We thank the reviewer for the suggestion. In the revised manuscript, the identifications of iron and $C_nX_mCOO^-$ -based light-harvesting species have been convincingly demonstrated (Fig. 2F, 3A, 3B and 3C, Fig. S30, S40, S41, S43, S49 and S50), which would significantly improve the influence of this protocol. We are now looking forward to the final positive recommendation from the reviewer.

Other minor points:

5.- In page 5, lines 138-143, the terms “benzoyl” and “sulfonyl” are confusing. It would be better to use the terms “benzoate” and “sulfonate”.

Our Response: We thank the reviewer for the suggestion. We have corrected “benzoyl” and “sulfonyl” into “benzoate” and “sulfonate” in the revised manuscript.

6.- In page 6, lines 147-148, the authors refers to “a-methyl and internal olefins ... that yields the diverse alkyl fluorides 39-41”. However, in the corresponding Figure 3, only one a-methyl olefin is functionalized (41), and no example of internal olefin is shown in such Figure. The main text must be corrected accordingly.

Our Response: We thank the reviewer for pointing out this mistake, which have now been corrected in the revised version.

7.- In Figure 5, reaction conditions for the functionalized products 58-62 derived from Febuxostat are missing.

Our Response: We thank the reviewer for pointing out this mistake, which have now been corrected in the revised version (Fig. 6).

Response to the comments from Reviewer #3.

The manuscript by Jiang et. al. describes the photocatalytic activation of trifluoroacetate and other haloalkylcarboxylates that is attributed to dissociative LMCT excitation of in situ generated Fe(III) complexes in the presence of Brønsted acids. In combination with a fluorine radical source, fluoro-polyhaloalkylated products can be obtained from non-activated alkenes including pharmaceutically relevant structures.

Photocatalytic reactions exploiting dissociative LMCT for organic synthesis are attracting significant interest and I can imagine that the present work makes a valuable addition to the increasing repertoire of synthetic protocols.

Our Response: We thank the reviewer for the positive comments on our work.

Conceptually, the present work is however very similar to other photocatalytic protocols based on the generation of various radicals via dissociative LMCT. For the specific challenge of photochemical CF₃COOH activation, alternative approaches have been previously reported (Stephenson, Nat Commun 6, 7919 (2015)) and a recent preprint (DOI: 10.26434/chemrxiv-2023-j1bwd) also describes a case of CF₃COOH activation that is supposed to proceed via LMCT excitation of in situ generated Fe(III) complexes.

Our Response: We thank the reviewer for the comments and suggestions. We appreciate the reported approaches for the activation of trichloroacetates to CF₃ radical, which encourage us to disclose the deeper mechanistic insights of acid unlocking sustainable iron photocatalysis and develop valuable fluoro-polyhaloalkylation of non-activated alkenes (Fig. 2F, 3A, 3B, 3C and 5B, Fig. S30, S40, S41, S43, S49 and S50).

In this work, the authors emphasize very much the importance of Brønsted acid for the photoreaction. In this regard, a major shortcoming is the lacking characterization of the photoactive species and the role of the Brønsted acid for its reactivity. The available data is limited to UV-vis spectra (Fig. 2E) and does not provide much insight. Consequently, the characterization of the photoactive species as "Fe(CF₃COO)₃ combined with acids" that are "unlocking" the LMCT reaction remains much too vague in my view.

Our Response: We thank the reviewer for the comments and suggestions. In the revised manuscript, we have supplemented the detailed mechanistic studies including DFT calculations, UV-Vis and ESI-HRMS experiments, etc., evidencing that the real iron-based light-harvesting species under blue light irradiation may derive from the in situ-assembly of Fe³⁺, C_nX_mCOO⁻, H⁺ and solvent acetonitrile, in which the effect of Brønsted acid via the hydrogen-bond interaction indeed increase the efficiency of LMCT between iron center and C_nX_mCOO⁻ (Fig. 2F, 3A, 3B and 3C, Fig. S30, S40, S41, S43, S49 and S50).

It also remains unclear to me why haloalkylcarboxylates and their conjugate acid are not

added directly but have to be provided by the “construction of a balanced system” (Fig. 2A, B).

Our Response: We thank the reviewer for the comments. As we can see (Fig. 2D, Fig. S21), compared to the direct TFA system, the faster initial reaction rate and shorter induction period of our standard conditions evidence the necessity and advantages of our balanced systems of Brønsted acid and ligand $C_nX_mCOO^-$ in the revised manuscript.

In summary, I can imagine that the protocol reported in this manuscript might be interesting to a more specialized audience. I am however convinced that more insight into the nature of the photoactive species and the role of the Bronsted acid would be necessary to make this work more suitable for publication in Nature Communications.

Our Response: We thank the reviewer for the comments and suggestions. In the revised manuscript, we have provided more insights into the nature of the iron based photoactive species and the role of the Bronsted acid through more detailed mechanistic studies (Fig. 2F, 3A, 3B and 3C, Fig. S30, S40, S41, S43, S49 and S50). During the revision of our manuscript, West and Xia published the similar Bronsted acid unlocking iron LMCT strategy for the hydrofluoroalkylation of alkenes (*Nat. Chem.* **2023**, *15*, 1683; *ACS Catal.* **2024**, *14*, 1300). Moreover, Nocera also demonstrated the activation of perfluoroalkyl carboxylates to C_nF_m radical by Ag-based LMCT under blue light irradiation (*Science* **2024**, *383*, 279). These elegant reports indicate metal-based photolysis via LMCT for the activation of inert compounds is highly significant. After the revision, we have disclosed that the real iron-based light-harvesting species under blue light irradiation may derive from the in situ-assembly of Fe^{3+} , $C_nX_mCOO^-$, H^+ and solvent acetonitrile, in which the effect of Brønsted acid via the hydrogen-bond interaction indeed increase the efficiency of LMCT between iron center and $C_nX_mCOO^-$. The synthetic significance of fluoro-polyhaloalkylation of alkenes from inert haloalkylcarboxylate ($C_nX_mCOO^-$, X= F or Cl) and the important mechanistic insights into the assembly of acid unlocking iron and $C_nX_mCOO^-$ -based light harvesting species should support this manuscript for the publication in Nature Communications.

REVIEWER COMMENTS

Reviewer #1 (Remarks to the Author):

I think the authors have addressed the questions that I posed with sufficient evidence. While the mechanism is still not convincing, I think the synthetic utility makes the paper suitable for publication.

Reviewer #2 (Remarks to the Author):

See attached pdf

Reviewer #3 (Remarks to the Author):

In the revised manuscript by Jiang et. al. the authors describe DFT calculations and mass spectrometry measurements aiming at the identification of the photoactive species and the role of the Bronsted acid. These results support the possible formation of a monocationic ferric complex by coordination of acetonitrile and carboxylate ligands where H-bonding between carboxylates, alternatively between a carboxylate and fluoride ligand, is enabled by protonation. Computational absorption spectra suggest that the H-bonding motif is required for a low-lying LMCT state accessible by excitation with visible light.

The calculated relative stability of the proposed photoactive species over alternative structures is however moderate and the relevance of species formed under mass spec conditions for the catalytic reaction might be questioned. Compared to cases where the photocatalyst is an isolated species that can be characterized with regard to its molecular structure, spectroscopic properties and excited state reactivity (Nocera, Science 2024, 383, 279–284), the present work still suffers from a significant level of uncertainty regarding the underlying photochemistry.

Despite these remaining ambiguities, it is obvious that substantial efforts have been made to characterize the photoactive species that often remains elusive in studies of photocatalysis with in situ formation of the catalyst. While this could be seen as some progress over the state of the art, I cannot see that the insight gained from this study has the potential to affect this field of research in a way that adds new concepts or triggers new directions of research. I am therefore not fully convinced that this manuscript is of sufficiently general interest for the readership of Nature Communications.

Reviewer #4 (Remarks to the Author):

As a theoretical and computational chemist, I will primarily focus on the Density Functional Theory (DFT) calculations presented in this manuscript. The revised version integrates detailed mechanistic experiments with comprehensive quantum chemical calculations, significantly advancing our understanding of the role of iron catalysts and acids. Overall, I highly recommend publication of this revised manuscript in Nature Communications, contingent upon addressing the following issues:

1. Iron complexes, particularly those featuring weak field ligands as reported here, often exhibit multiple spin states with energies in close proximity. It is essential to inquire whether the authors thoroughly accounted for these different spin states when assessing the

potential of Ligand-to-Metal Charge Transfer (LMCT) and Single Electron Transfer (SET) events.

2. The authors articulated the role of intramolecular hydrogen bond in activating the iron complex and provided detailed frontier orbital analysis. To enrich the discussion on the influence of acid in facilitating LMCT processes, it would be beneficial to compare the orbital energies of the complex with and without intramolecular hydrogen bonding.

3. In Fig. 3B, it appears that 12' should be neutral in the absence of a proton. It is imperative to evaluate the thermodynamic change of SET for the complex where the hydrogen bond is inaccessible, providing a comparison to the SET process in 12.

The revised manuscript by Jiang et al. has been notably improved from previous version. Regarding my initial comments, the authors have provided reasonable answers to most of the points of concern, and following the reviewer's suggestions, the authors have carried out several experiments to elucidate the role of the different components of the reaction (CH₃CN, oxydibenzene, iPrOH, selectfluor), thus shedding some light to the real nature of the iron species implicated in the photocatalytic cycle. Thanks to this mechanistic study, the authors were able to identify two different iron(III) species **12** and **87** by using ESI-HRMS spectrometry, and DFT investigations pointed to the plausible assistance in **12** of H-bonding interactions to enable and facilitate the Fe(III)-O₂CCF₃ homolytic bond scission. The following CO₂ elimination produces CF₃ radicals (or the analogous C_nF_m radicals) that are trapped by the olefin and selectfluor yielding the fluorotrifluoromethylated products. The obscure mechanistic scenario previously proposed by the authors, is now (partially) clarified, and my main point of concern does not apply anymore. Accordingly, this reviewer would like to thank the authors for the efforts carried out to address all the points argued by the reviewers, and based on the quality of the herein reported results in medicinal and fluorine chemistry (also corroborated by the recent acceptance of closely related works in Top Ranked journals (F. Hernandez-Julia et al. *Angew. Chem. Int. Ed.* **2024**, *63*, e202311984; J. G. West et al. *Nat. Chem.* **2023**, *15*, 1683; W. Xia et al. *ACS Catal.* **2024**, *14*, 1300)), I am pleased to recommend publication of this work in *Nature Communications* once the following minor points are properly addressed:

Points for the revision:

- 1.-** Figure 3b and 3d shows mechanistic considerations and the most plausible pathway by which this transformation takes place according to DFT calculations. If I am not wrong, the calculated values only refer to the feasibility of those elementary steps in terms of thermodynamics but not in terms of kinetics. A more complete picture of the calculated pathway should be illustrated (at least in the ESI) indicating and discussing the feasibility of the trifluoroacetate radical release from **12** (and **87**) both in terms of kinetics and thermodynamics. Otherwise, such an investigation is not conclusive.
- 2.-** Regarding the iron(III) fluoride complex **87** detected by ESI-HRMS spectrometry, and related to the previous point, which is the role of **87** in the photocatalytic reaction? According to Figure 3d, only **12** participates in the catalytic reaction. In other words, is **87** a catalytic intermediate for this transformation? Or alternatively, is it an unwanted decomposition product that contributes to the catalyst deactivation? This remains unclear and must be clarified in the manuscript. Probably, DFT calculations could be useful here.
- 3.-** Page 6, line 169, what does it mean the work "impressible" in such a context?
- 4.-** Some passages of the main text remain unclear and difficult to follow. The manuscript would benefit from an additional round of proofreading, if possible by an English native speaker, making the reading easier and making this work more accessible to a larger audience.

Response to the comments from Reviewer #1.

I think the authors have addressed the questions that I posed with sufficient evidence. While the mechanism is still not convincing, I think the synthetic utility makes the paper suitable for publication.

Our Response: We sincerely thank the reviewer for the positive comments on our work. Both Brønsted acid and CH₃CN are indeed necessary to assemble the iron and C_nX_mCOO⁻-based light-harvesting species for C_nX_m radical production (Figs. 2F, S39, and 3B, X = F or Cl). In the second revised manuscript, a complete picture of the calculated pathway for mechanism was illustrated to confirm the reasonability of current mechanism (Figs. S59 and S60). Additionally, our further efforts in related study using similar Bronsted acid-unlocked iron LMCT strategy are currently underway. Thank you again for your strong support for our work.

Fig. 2F UV-vis experiments.

Fig. S39 Acetonitrile loading experiments.

Fig. 3B Density functional theory (DFT) calculations.

Fig. S59 DFT calculation for SET process.

Fig. S60 Free energy profile for the generation of the product.

Response to the comments from Reviewer #2.

The revised manuscript by Jiang et al. has been notably improved from previous version. Regarding my initial comments, the authors have provided reasonable answers to most of the points of concern, and following the reviewer's suggestions, the authors have carried out several experiments to elucidate the role of the different components of the reaction (CH₃CN, oxydibenzene, iPrOH, selectfluor), thus shedding some light to the real nature of the iron species implicated in the photocatalytic cycle. Thanks to this mechanistic study, the authors were able to identify two different iron(III) species 12 and 87 by using ESI-HRMS spectrometry, and DFT investigations pointed to the plausible assistance in 12 of H-bonding interactions to enable and facilitate the Fe(III)-O₂CCF₃ homolytic bond scission. The following CO₂ elimination produces CF₃ radicals (or the analogous C_nF_m radicals) that are trapped by the olefin and selectfluor yielding the fluorotrifluoromethylated products. The obscure mechanistic scenario previously proposed by the authors, is now (partially) clarified, and my main point of concern does not apply anymore. Accordingly, this reviewer would like to thank the authors for the efforts carried out to address all the points argued by the reviewers, and based on the quality of the herein reported results in medicinal and fluorine chemistry (also corroborated by the recent acceptance of closely related works in Top Ranked journals (F. Hernandez-Julia et al. Angew. Chem. Int. Ed. 2024, 63, e202311984; J. G. West et al. Nat. Chem. 2023, 15, 1683; W. Xia et al. ACS Catal. 2024, 14, 1300)), I am pleased to recommend publication of this work in Nature Communications once the following minor points are properly addressed:

Our Response: Thank you very much for your supportive and insightful comments regarding our manuscript. We are deeply grateful for your recognition of the significance of our work in the field and your support for its publication in Nature Communications.

1.- Figure 3b and 3d shows mechanistic considerations and the most plausible pathway by which this transformation takes place according to DFT calculations. If I am not wrong, the calculated values only refer to the feasibility of those elementary steps in terms of thermodynamics but not in terms of kinetics. A more complete picture of the calculated pathway should be illustrated (at least in the ESI) indicating and discussing the feasibility of the trifluoroacetate radical release from 12 (and 87) both in terms of kinetics and thermodynamics. Otherwise, such an investigation is not conclusive.

Our Response: We thank the reviewer for the suggestions and comments. In the revised supplementary information, a complete picture of the calculated pathway for mechanism has been illustrated in **Figs. S59 and S60** to showcase the reasonability of current mechanism including the feasibility of the trifluoroacetate radical release from excited **12**. In the next answer, we have also discussed the role of **87**.

Fig. S59 DFT calculation for SET process.

Fig. S60 Free energy profile for the generation of the product.

2.- Regarding the iron(III) fluoride complex 87 detected by ESI-HRMS spectrometry, and related to the previous point, which is the role of 87 in the photocatalytic reaction? According to Figure

3d, only **12** participates in the catalytic reaction. In other words, is **87** a catalytic intermediate for this transformation? Or alternatively, is it an unwanted decomposition product that contributes to the catalyst deactivation? This remains unclear and must be clarified in the manuscript. Probably, DFT calculations could be useful here.

Our Response: We thank the reviewer for the comments and suggestions. As we can see, the formation of **87** require the F^- generation from Selectfluor, while Selectfluor plays the role as an electrophilic fluorination reagent to capture the alkyl radical intermediate **15** in this system. Considering the fact that exogenous F^- shows the suppression effect for alkene fluorotrifluoromethylation (Fig. S57), we proposed that the concentration of F^- under the standard conditions should be low. Although **87** was detected by ESI-HRMS spectrometry and DFT calculations also demonstrated its feasibility of generating CF_3 radical under blue light irradiation (Fig. S52), the low concentration of F^- in the process of reaction may indicate that **87** is not the first choice in comparison to **12**, but alternative. To further understand the role of possible species **87**, this protocol was extended to the trifluoromethylation of arene though directly replacing Selectfluor into sodium persulfate. As shown in Fig. S56, even in the absence of fluorine source, this protocol is still feasible to produce CF_3 radical, while the exogenous F^- do not promote the trifluoromethylation of arene. These results indicate that **87** is not necessary for this protocol and **12** is the first choice for effective iron-based light-harvesting species under this standard conditions.

Fig. S57 Investigation of exogenous F^- effect for fluorotrifluoromethylation of alkenes.

Fig. S52 DFT calculation of **87**.

Entry	TBAF (x mol%)	103, Yield
1	0	22%
2	40	9%
3	100	6%

Fig. S56 Investigation of exogenous F⁻ effect for trifluoromethylation of arenes.

3.- Page 6, line 169, what does it mean the work “impressible” in such a context?

Our Response: We thank the reviewer for the comment. We have corrected this word into “susceptible to...” in the revised manuscript.

4.- Some passages of the main text remain unclear and difficult to follow. The manuscript would benefit from an additional round of proofreading, if possible by an English native speaker, making the reading easier and making this work more accessible to a larger audience.

Our Response: We thank the reviewer for the suggestions. In the revised manuscript, we have carefully corrected the language with the help of an English native speaker. Many thanks for your strong support for our work.

Response to the comments from Reviewer #3.

In the revised manuscript by Jiang *et. al.* the authors describe DFT calculations and mass spectrometry measurements aiming at the identification of the photoactive species and the role of the Bronsted acid. These results support the possible formation of a monocationic ferric complex by coordination of acetonitrile and carboxylate ligands where H-bonding between carboxylates, alternatively between a carboxylate and fluoride ligand, is enabled by protonation. Computational absorption spectra suggest that the H-bonding motif is required for a low-lying LMCT state accessible by excitation with visible light.

Our Response: We thank the reviewer for the positive comments on our work.

The calculated relative stability of the proposed photoactive species over alternative structures is however moderate and the relevance of species formed under mass spec conditions for the catalytic reaction might be questioned. Compared to cases where the photocatalyst is an isolated species that can be characterized with regard to its molecular structure, spectroscopic properties and excited state reactivity (Nocera, *Science* 2024, 383, 279–284), the present work still suffers from a significant level of uncertainty regarding the underlying photochemistry.

Our Response: We thank the reviewer for the comments to improve our work. During the first revision of our manuscript, Prof. West published the similar Bronsted acid-unlocked iron LMCT strategy for the hydrofluoroalkylation of alkenes (*Nat. Chem.* 2023, 15, 1683), in which there is no study on the determination of iron-based light-harvesting species. To promote the development of iron photocatalysis, we have carried out detailed mechanistic studies aiming to reveal the possibly real and effective structure of this iron-based light-harvesting species. As you can see, Brønsted acid and CH₃CN are indeed necessary to assemble the iron and C_nX_mCOO⁻-based light-harvesting species **12** for C_nX_m radical production (Figs. 2E and 2F, 3A-3C, and S40-S50). Compared to Prof. Nocera's [Ag(bpy)₂(TFA)][OTf] and Ag(bpy)(TFA)₂, our **12** is very difficult to be isolated due to its structural instability. We have tried our best to isolate **12** but failed. Comparatively speaking, our work also shows the attractive sustainability and utility of iron-based photochemistry. In a follow-up study, we are using various bidentate/tridentate ligands to stabilize this iron-based light-harvesting species to disclose deeper insights of iron photocatalysis.

Fig. 3B Density functional theory (DFT) calculations.

Despite these remaining ambiguities, it is obvious that substantial efforts have been made to characterize the photoactive species that often remains elusive in studies of photocatalysis with in

situ formation of the catalyst. While this could be seen as some progress over the state of the art, I cannot see that the insight gained from this study has the potential to affect this field of research in a way that ads new concepts or triggers new directions of research. I am therefore not fully convinced that this manuscript is of sufficiently general interest for the readership of Nature Communications.

Our Response: We thank the reviewer for the comments and suggestions. Due to the following unique advantages of this work, we believe that our work has the novelty, significance, and broad interest, which merits its publication as an article in Nature Communications:

- (1) Robust Brønsted acid-unlocked iron ligand-to-metal charge transfer (LMCT) catalytic platform for fluoro-polyhaloalkylation of non-activated alkenes.
- (2) Utilization of the inert and abundant haloalkylcarboxylate as the real haloalkyl radical source.
- (3) Wide scope, late-stage functionalization, gram scale synthesis, 160 TON of iron catalyst, providing a practical route toward drug candidates.
- (4) Detailed mechanistic studies and significant insight: in situ-assembly of Fe^{3+} , $\text{C}_n\text{X}_m\text{COO}^-$, H^+ , and acetonitrile solvent to form iron-based light-harvesting species, without involving noble metal and complex/unobtainable ligand. This encourages synthetic chemists to develop more practical and sustainable 3d metal-based photocatalysis.

Response to the comments from Reviewer #4.

As a theoretical and computational chemist, I will primarily focus on the Density Functional Theory (DFT) calculations presented in this manuscript. The revised version integrates detailed mechanistic experiments with comprehensive quantum chemical calculations, significantly advancing our understanding of the role of iron catalysts and acids. Overall, I highly recommend publication of this revised manuscript in *Nature Communications*, contingent upon addressing the following issues:

Our Response: Thank you very much for your supportive and insightful comments regarding our manuscript. We are deeply grateful for your recognition of the significance of our work in the field and your support for its publication in *Nature Communications*.

1. Iron complexes, particularly those featuring weak field ligands as reported here, often exhibit multiple spin states with energies in close proximity. It is essential to inquire whether the authors thoroughly accounted for these different spin states when assessing the potential of Ligand-to-Metal Charge Transfer (LMCT) and Single Electron Transfer (SET) events.

Our Response: We thank the reviewer for the comments and suggestions. In our catalytic cycle, iron-based light-harvesting species **12** can undergo Ligand-to-Metal Charge Transfer (LMCT) under blue light irradiation. The Single Electron Transfer (SET) between iron(II) species **14** and radical cations **10** or **17** is responsible for the recycling of iron(III) (Figs. R1 and S59). We have carefully considered different spin states of **12** and **14**. As we can see, high spin is the most stable state. Moreover, there is no coexistence of multiple spin states (Fig. S51).

Fig. R1 Catalytic cycle of **12**.

Fig. S59 DFT calculation for SET process.

Species	G (a.u.)
12 (doublet)	-3108.130262
12 (quartet)	-3108.137950
12 (hextet)	-3108.161111
14 (singlet)	-3108.325434
14 (triplet)	-3108.350736
14 (quintet)	-3108.376685

Fig. S51 Spin states determination of 12 and 14.

2. The authors articulated the role of intramolecular hydrogen bond in activating the iron complex and provided detailed frontier orbital analysis. To enrich the discussion on the influence of acid in facilitating LMCT processes, it would be beneficial to compare the orbital energies of the complex with and without intramolecular hydrogen bonding.

Our Response: We thank the reviewer for the comments and suggestions. In the revised supplementary information, we have compared the orbital energies of 12 and 12' (Figs. R2 and S50). In the presence of intramolecular hydrogen bonding, the LUMO orbital energy of iron significantly decreased by 0.86 eV, which is beneficial to the LMCT process.

Fig. R2 Structure of **12** and **12'**.

Species	E (a.u.)	Electron volt (eV)
12 (LUMO)	-0.1892	-5.1462
12' (LUMO)	-0.1575	-4.2840

Fig. S50 LUMO energy comparison between **12** and **12'**.

3. In Fig. 3B, it appears that **12'** should be neutral in the absence of a proton.

Our Response: We thank the reviewer for pointing out this mistake, which have now been corrected in the revised version (Fig. 3B).

It is imperative to evaluate the thermodynamic change of SET for the complex where the hydrogen bond is inaccessible, providing a comparison to the SET process in **12**.

Our Response: We thank the reviewer for the comments and suggestions. Actually, in the manuscript, we have provided the thermodynamic comparisons when excited **12** and **12'** release $\text{CF}_3\text{COO}^\bullet$ radical and corresponding iron(II) intermediates (Fig. 3B). The results indicate that the intramolecular hydrogen bonding effectively hinders the recombination of iron(II) and $\text{CF}_3\text{COO}^\bullet$ radical, thus, ensuring the subsequent decarboxylation to CF_3 radical on the rails (Fig. S60). We have also evaluated and compared the SET between Fe^{3+} and CF_3COO^- of **12** and **12'** under darkness (Fig. S49). The calculated results show that Brønsted acid-mediated hydrogen bonding can increase the oxidizing ability of iron(III) center.

Fig. 3B Density functional theory (DFT) calculations.

Fig. S49 SET comparison between **12** and **12'** without light irradiation.

Fig. S60 Free energy profile for the generation of the product.

REVIEWERS' COMMENTS

Reviewer #2 (Remarks to the Author):

See attached document

Reviewer #4 (Remarks to the Author):

I think all the scientific issues I raised have been fully addressed and the manuscript is suitable for the publication now.

The manuscript has been improved again compared to the previous version, including by the polishing of the English language that makes now a more pleasant reading. The authors have addressed all my previous points, and I am mostly satisfied by their answers, although I keep thinking (and intrigued) about the real role of the fluoro-Fe^{III} complex **87** along this transformation. Indeed, DFT calculations indicate that the reduction to Fe^{II} from **87** with the concomitant O₂CCF₃ radical liberation (lately producing the CF₃ radical via CO₂ extrusion) is significantly more exergonic than the analogous process from **12**. Thus, this reviewer believes that the role of **87** may be eventually important in the current transformation, and that **12** and **87** cooperatively produce the desired compounds. Anyhow, as mentioned in my previous recommendation during the 2nd round of peer-review, according to the high quality and potential relevance of the selective fluoro-trifluoromethylation of olefins (and the related transformations herein reported) in medicinal and fluorine chemistry, along with the coherent mechanistic investigation that clarified the role of the Brønsted acid to enable the CF₃ radical release, I firmly believe that it is the time now for publication of the current manuscript in *Nature Communications* and let it go for the scrutiny of the readers and colleagues working in the field.

Some additional minor comments and suggestions of modifications are indicated below:

- 1.-** In Fig. 1A, the [O] symbol should be well centered with the arrow to avoid confusion. It took me some time to understand why the metal becomes oxidized upon coordination of L⁻ until I found the [O] symbol. Additionally, there is a typo on bottom of Fig. 1A: it should read as “a) Overcoming the limitation of redox potential” instead of “a) Overcoming the limitation of redox poential”
- 2.-** In the same Fig. 1A, below the L = CF₃COO⁻, I would suggest to highlight that this is “scarcely studied” and “mechanistically challenging” instead of the current description as “unknown mechanistic science”. Actually, although there are only a few examples on the field, others have already studied the efficient CF₃ radical release from trifluoroacetate ligands using Fe promoters and light (see for instance the work of Juliá-Hernández and co-workers, ref. 66).
- 3.-** In page 7, lines 10-11, the authors mention that “Not only were terminal olefins feasible, but also α- methyl olefins were successfully functionalized to enrich the diversity of alkyl fluorides (**43-45**).” Only products **43** and **45** contain a methyl substituent in the olefin; compound **44** is formed from a monosubstituted olefin. In addition, these substrates containing a secondary olefin keep being terminal ones, and thus, that sentence is confusing. In fact, the use of an internal olefin would be very interesting here, as in such a case example, the initial addition of the CF₃ radical may display regioselectivity issues. In the absence of such an experiment, to avoid confusion, I would suggest the authors to rephrase the above mentioned sentence.

Response to the comments from Reviewer #2.

*The manuscript has been improved again compared to the previous version, including by the polishing of the English language that makes now a more pleasant reading. The authors have addressed all my previous points, and I am mostly satisfied by their answers, although I keep thinking (and intrigued) about the real role of the fluoro-Fe^{III} complex **87** along this transformation. Indeed, DFT calculations indicate that the reduction to Fe^{II} from **87** with the concomitant O₂CCF₃ radical liberation (lately producing the CF₃ radical via CO₂ extrusion) is significantly more exergonic than the analogous process from **12**. Thus, this reviewer believes that the role of **87** may be eventually important in the current transformation, and that **12** and **87** cooperatively produce the desired compounds. Anyhow, as mentioned in my previous recommendation during the 2nd round of peerreview, according to the high quality and potential relevance of the selective fluoro-trifluoromethylation of olefins (and the related transformations herein reported) in medicinal and fluorine chemistry, along with the coherent mechanistic investigation that clarified the role of the Brønsted acid to enable the CF₃ radical release, I firmly believe that it is the time now for publication of the current manuscript in Nature Communications and let it go for the scrutiny of the readers and colleagues working in the field.*

Our Response: Thank you very much for your supportive and insightful comments regarding our manuscript. We are deeply grateful for your recognition of the significance of our work in the field and your support for its publication in Nature Communications.

Response to the comments from Reviewer #2.

1.- In Fig. 1A, the [O] symbol should be well centered with the arrow to avoid confusion. It took me some time to understand why the metal becomes oxidized upon coordination of L until I found the [O] symbol.

Our Response: We thank the reviewer for the suggestion. In the revised manuscript, we have corrected the position of the [O] symbol to avoid confusion (see Page 11, Figure 1a).

Additionally, there is a typo on bottom of Fig. 1A: it should read as “a) Overcoming the limitation of redox potential” instead of “a) Overcoming the limitation of redox poential”.

Our Response: We sincerely thank the reviewer for pointing out this issue. We have corrected “Overcoming the limitation of redox poential” into “Overcoming the limitation of redox potential” in the revised manuscript (see Page 11, Figure 1a).

2.- In the same Fig. 1A, below the L = CF₃COO⁻, I would suggest to highlight that this is “scarcely studied” and “mechanistically challenging” instead of the current description as “unknown mechanistic science”. Actually, although there are only a few examples on the field, others have already studied the efficient CF₃ radical release from trifluoroacetate ligands using Fe promoters and light (see for instance the work of Juliá-Hernández and co-workers, ref. 66).

Our Response: We sincerely thank the reviewer for pointing out this mistake. We have corrected

“unknown mechanistic science” into “scarcely studied” and “mechanistically challenging” in the revised manuscript (see Page 11, Figure 1a).

3.- In page 7, lines 10-11, the authors mention that “Not only were terminal olefins feasible, but also α -methyl olefins were successfully functionalized to enrich the diversity of alkyl fluorides (43-45).” Only products 43 and 45 contain a methyl substituent in the olefin; compound 44 is formed from a monosubstituted olefin. In addition, these substrates containing a secondary olefin keep being terminal ones, and thus, that sentence is confusing. In fact, the use of an internal olefin would be very interesting here, as in such a case example, the initial addition of the CF_3 radical may display regioselectivity issues. In the absence of such an experiment, to avoid confusion, I would suggest the authors to rephrase the above mentioned sentence.

Our Response: We thank the reviewer for the comment. We have corrected the description *“Not only were terminal olefins feasible, but also α -methyl olefins were successfully functionalized to enrich the diversity of alkyl fluorides (43-45).” into “The allylbenzene derivative and 1-hexadecene were also feasible (43 and 44). Additionally, α,α -disubstituted olefin was successfully functionalized to enrich the diversity of alkyl fluorides (45).” (see Page 4, the third paragraph).*

Response to the comments from Reviewer #4.

I think all the scientific issues I raised have been fully addressed and the manuscript is suitable for the publication now.

Our Response: Thank you very much for your supportive comments regarding our manuscript. We are deeply grateful for your recognition of the significance of our work in the field and your support for its publication in Nature Communications.